# A renewably sourced, circular photopolymer resin for additive manufacturing

Thiago O. Machado[1,3], Connor J. Stubbs[1,3], Viviane Chiaradia[1], Maher A. Alraddadi[1], Arianna Brandolese[1], Joshua C. Worch[1,2 ✉] & Andrew P. Dove[1 ✉]

The additive manufacturing of photopolymer resins by means of vat photopolymerization enables the rapid fabrication of bespoke 3D-printed parts. Advances in methodology have continually improved resolution and manufacturing speed, yet both the process design and resin technology have remained largely consistent since its inception in the 1980s[1]. Liquid resin formulations, which are composed of reactive monomers and/or oligomers containing (meth)acrylates and epoxides, rapidly photopolymerize to create crosslinked polymer networks on exposure to a light stimulus in the presence of a photoinitiator[2]. These resin components are mostly obtained from petroleum feedstocks, although recent progress has been made through the derivatization of renewable biomass[3–6] and the introduction of hydrolytically degradable bonds[7–9]. However, the resulting materials are still akin to conventional crosslinked rubbers and thermosets, thus limiting the recyclability of printed parts. At present, no existing photopolymer resin can be depolymerized and directly re-used in a circular, closed-loop pathway. Here we describe a photopolymer resin platform derived entirely from renewable lipoates that can be 3D-printed into high-resolution parts, efficiently deconstructed and subsequently reprinted in a circular manner. Previous inefficiencies with methods using internal dynamic covalent bonds[10–17] to recycle and reprint 3D-printed photopolymers are resolved by exchanging conventional (meth)acrylates for dynamic cyclic disulfide species in lipoates. The lipoate resin platform is highly modular, whereby the composition and network architecture can be tuned to access printed materials with varied thermal and mechanical properties that are comparable to several commercial acrylic resins.

State-of-the-art photocurable dynamic networks can be reformulated to reprint or recure by the addition of extra photoactive resin components in an open-loop process[15–19], however, the orthogonality between the dynamic bonds and the photochemical crosslinking reactions leads to regenerated materials that have a distinct composition (compared to the pristine sample). Moreover, the requirement for additional reactive species at each successive recycling step results in a 'snowballing' effect, in which the only way to recycle the material is to make more. To realize a circular photocurable network, the dynamic bond must be formed by photopolymerization, that is, in situ during the crosslinking, and the network must depolymerize back into the original units so that it can be repeatedly photocured or printed (Fig. 1c). Whereas two-dimensional (2D)-photoset materials based on reversible cycloaddition reactions[20–23] and thiol-ene[24] chemistry partially satisfy this requirement, their translation to high-fidelity vat photopolymerization three-dimensional (3D) printing at scale is beset by several issues including the non-trivial syntheses of the resin components, the need for high-energy and/or continuous light irradiation, incomplete depolymerization or substantial disparity in properties after

recycling. On account of their ready polymerization by radical mediated methods[25–27] and established dynamic behaviour[28], including depolymerization[29–31], we proposed that applying strained cyclic disulfides, such as naturally sourced lipoic acid, would enable closed-loop vat photopolymerization printing by maintaining a concentration of disulfide in the resin that was sufficiently high to allow rapid curing without the need for additives that would make the material irreversible, but not so high as to render the resin unstable (Fig. 1c)[29].

Lipoic acid was esterified using a one-step (1-ethyl-3-(3-dimethylaminopropyl)carbodiimide) EDC coupling[32] with renewably sourced isosorbide and menthol to form isosorbide lipoate (IsoLp$_2$), a multivalent crosslinker and menthyl lipoate (MenLp$_1$), a reactive diluent. Although the use of EDC and chlorinated solvents create environmental, health and safety challenges for monomer synthesis, there are recent coupling protocols[33] using less-toxic reagents and solvents that could be implemented. Good ambient stability was observed for both lipoates when diluted in solvent; however, on concentration, IsoLp$_2$ was susceptible to gelation on prolonged storage, probably because

[1]School of Chemistry, University of Birmingham, Edgbaston, Birmingham, UK. [2]Department of Chemistry, Macromolecules Innovation Institute, Blacksburg, VA, USA. [3]These authors contributed equally: Thiago O. Machado, Connor J. Stubbs. ✉e-mail: jworch@vt.edu; a.dove@bham.ac.uk

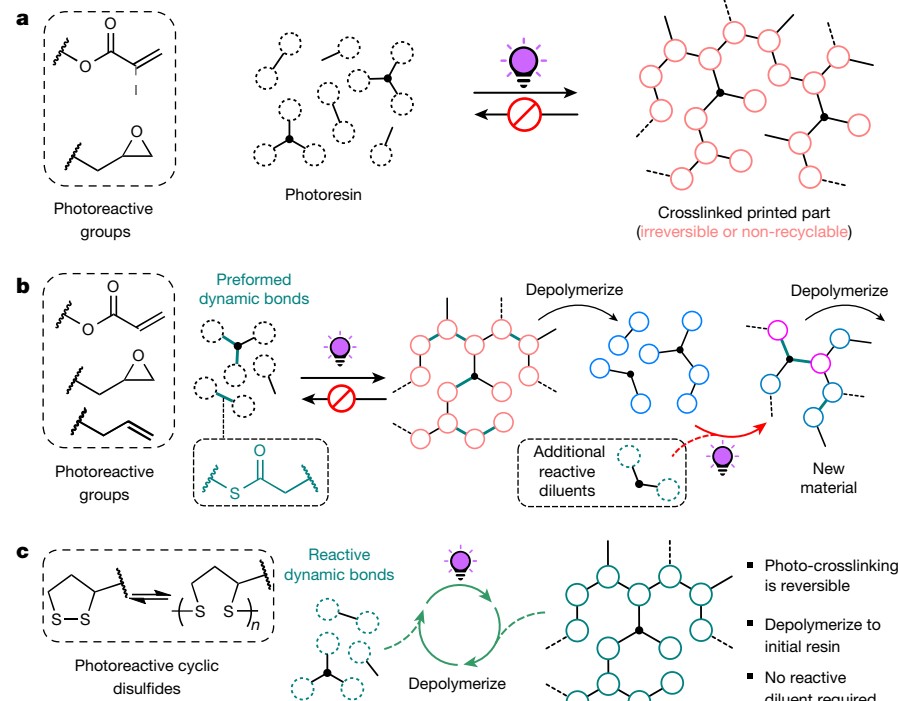

**Fig. 1 | 3D printing of photoresins and their recycling. a**, Conventional resins for 3D printing: irreversible photoinduced crosslinking of typical photoresins for additive manufacturing by stereolithography (SLA) or Digital Light Processing (DLP) using acrylate or epoxy functional (macro)monomers. **b**, Open-loop recycling in 3D printing uses preformed dynamic bonds in photoset materials to enable resin recycling after depolymerization and addition of reactive diluents to create new photocurable material. **c**, Closed-loop recycling in 3D printing (this work) presents a method to achieve polymerization–depolymerization cycles of dynamic disulfide bonds that enables the creation of renewably sourced resins suitable for closed-loop chemical recycling.

of premature self-crosslinking[34]. However, when both lipoate components were combined to form the resin, MenLp$_1$–IsoLp$_2$ (30:70 wt%), we observed that the mixture was comparatively more stable than either component in isolation (Fig. 2a), possibly a consequence of each mixed component serving as a diluent to the other, despite also possessing cyclic disulfide motifs. Nevertheless, it is advantageous for the practical translation of this resin system. To increase the greenness of the resin synthesis, we also demonstrated the synthesis of these components using an acid-catalysed Fisher esterification in bulk (that is, without solvent) (Supplementary Table 1 and Supplementary Figs. 13–15), which is commonly used during the manufacture of bulk commodity chemicals[35]. Partial gelation was observed for IsoLp$_2$ on cooling of the reaction mixture, which is probably due to thermal-assisted polymerization[36,37]. However, this was avoided when both alcohols were reacted simultaneously with lipoic acid to directly obtain MenLp$_1$–IsoLp$_2$ (28:72 wt%), as the mixed lipoate product was observed to be more stable than either constituent component. The formulated resin could be stored on the bench top when protected from light for several weeks while maintaining a consistent viscosity and rapid photocurability (Supplementary Fig. 165). It is important to note that ageing effects can be neutralized by depolymerizing back to monomer components (vide infra).

The MenLp$_1$–IsoLp$_2$ (30:70 wt%) resin can be cured with or without additives, that is, a photoinitiator, an opaquing agent or other reactive species, that would render the resultant material non-depolymerizable, such as acrylates[25,38]. Printing the MenLp$_1$–IsoLp$_2$ (30:70 wt%) resin on a commercial digital light processing (DLP) printer was achieved using a $\lambda$ = 405 nm light source (Supplementary Fig. 184). To assess printing resolution and fidelity, we designed a rectangular base containing several square arrays and bridges, which was inspired by methods from Page and coworkers[39] (Fig. 2b). The smallest feature we could reproducibly print was a 100 µm wall (roughly 3 pixels wide) at 25 s per 50 µm, highlighting the impressive $x$–$y$ resolution of our resin platform on an off-the-shelf commercial 3D printer. Furthermore,

the successful printing of bridge structures was illustrative of good resolution along the $z$ axis where only modest cure-through (113 ± 7% at 20 s curing) was observed for an over-hanging bridge (Fig. 2b). Commercial acrylic resins routinely incorporate opaquing agents to inhibit cure-through that can distort $z$-resolution (overcure greater than or equal to 200–300%) in the absence of an added opaquing agent[40]. It is likely that the $z$ axis resolution for the lipoate-based resins could be further optimized with the addition of an opaquing agent. The $x$–$y$ resolution of the print was also outstanding as evidenced by excellent agreement between the theoretical surface area and printed area of each square (Fig. 2c). This is particularly advantageous from a user-standpoint. We also successfully printed the resin to afford a high-resolution 3D part, '3DBenchy', that possessed challenging geometrical features, including over-hanging surfaces and holes that are difficult to build with conventional manufacturing methods (Fig. 2d). The high-fidelity, complex prints were obtained using 35 s cure per layer, which corresponds to a build rate of 5.1 mm h$^{-1}$ (excluding the peeling time process).

To investigate the chemical depolymerization of the printed lipoate network, base-catalysed depolymerization of the network formed from ring-opened polydisulfides was studied[30,32,34,36]. Placing the pulverized printed part in 2-methyl-tetrahydrofuran (MeTHF), a green solvent, and adding an equimolar combination of phosphazene (P$_1$-$t$-Bu):thiophenol at 1 mol% (relative to disulfide content) proved effective for this purpose (Fig. 2e). Other greener depolymerization methods were also screened and found to be effective (vide infra). The printed part was fully dissolved after heating at 80 °C for 3 h under a N$_2$ atmosphere and the recycled resin was isolated (see Supporting Information for details) in up to 98% yield to afford a viscous liquid (Fig. 2f). Size-exclusion chromatography (SEC) analysis of the recycled resin showed that it had a similar molecular weight profile to the original resin, although a small amount of oligomeric product was observed, which indicates that the depolymerization was not 100% effective (Fig. 2g). Nuclear magnetic

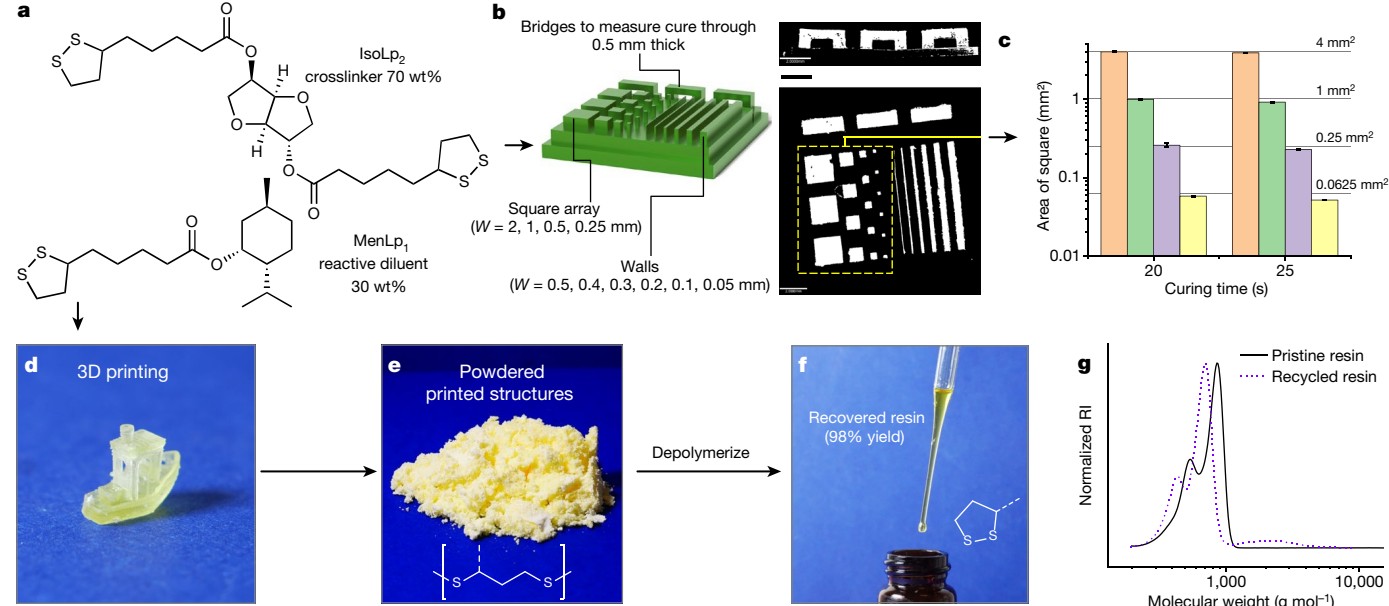

**Fig. 2 | 3D printing of MenLp₁–IsoLp₂ (30:70 wt%) and depolymerization of printed parts. a**, Chemical composition of formulated resin. **b**, High-resolution printing of 3D-printed part containing intricate square arrays and bridges. **c**, $x$–$y$ printing accuracy determined by comparing surface area of squares to curing time (pixel size 30 μm). Theoretical surface area of each square and number of squares (sample size): 4 mm² (n = 3); 1 mm² (n = 4); 0.25 mm² (n = 5); 0.0625 mm² (n = 6). Centre value is average surface area and error bars indicate 1 standard deviation. **d**, 3D-printed complex part. **e**, Photograph of powdered 3D-printed parts. **f**, Photograph of recovered resin from depolymerized 3D-printed parts, achieved in 98% yield. **g**, SEC of initial resin compared to recovered resin (CHCl₃ + 0.5% v/v NEt₃, against polystyrene standards, refractive index (RI) detector). Scale bar, 2 mm (**b**).

resonance (NMR) spectroscopic analysis of the recycled resin enabled calculation of the recycled resin purity of 85 mol%, with the minor component being an oligomeric species (Supplementary Fig. 30). The ratio of MenLp₁ to IsoLp₂ in the recycled resin was calculated to be a 1 to 1.53 mol ratio (31:69 wt%), which is within experimental error of the as-synthesized mixture before polymerization (printing). Together this indicates that the overall resin composition was comparable to the initial formulation, thus demonstrating that the resin was both 3D printable and could be depolymerized efficiently.

To explore the structure–property space of the platform, we first varied the ratio of reactive diluent to crosslinker in MenLp₁–IsoLp₂ (90:10 to 10:90 wt%) formulations (Fig. 3a,b). Each remained liquid at ambient temperature without the need for additional diluents. We created 2D photosets of roughly 0.5 mm thickness by curing them on glass slides (roughly 300–500 nm, 30 min irradiation, 1 wt% ethyl (2,4,6-trimethylbenzoyl) phenylphosphinate as a photoinitiator) to enable rapid screening of materials. The photoinitiator was incorporated to provide consistent and spatially even gelation of the resins at this thickness. These 2D photosets are suitable surrogates for approximating 3D-printed parts because the printed structures showed similar mechanical profiles, despite being slightly weaker overall (Supplementary Figs. 144 and 164). The ultimate tensile strength (UTS) and Young's modulus ($E$) were dependent on the crosslinker content with both values increasing with IsoLp₂ content (30 wt% IsoLp₂ UTS = 0.9 ± 0.1 MPa, $E$ = 0.014 ± 0.002 MPa; 90 wt% IsoLp₂ UTS = 15.6 ± 0.9 MPa, $E$ = 3.3 ± 0.2 MPa) (Fig. 3a). The glass transition temperature ($T_g$) also positively correlated to the crosslinker (IsoLp₂) amount and ranged from −30 to 20 °C (Fig. 3b).

The synthesis of a range of monomeric lipoates using other renewably sourced alcohols enabled us to probe the generality of the resin platform. Combination of IsoLp₂ with RLp₁ at a 30:70 wt% ratio (diluent to crosslinker) and subsequent photocuring formulations enabled us to create materials with a range of mechanical properties (Fig. 3c–f). Using an aromatic, guaiacol lipoate (GuaLp₁–IsoLp₂) in the resin resulted in a material with similar thermomechanical properties to the MenLp₁–IsoLp₂ resin; however, including an aliphatic lipoate (EtLp₁) resulted in a sample with decreased $T_g$ ($\Delta$ = 16 °C) and lower mechanical strength (Fig. 3d,e). Formulation of a resin synthesized from a long aliphatic alcohol (SteaLp₁–IsoLp₂) led to the observation of a melt-transition ($T_m$), although this value was near ambient temperature and the material was mechanically weak and brittle. All lipoate networks possessed adequate bulk thermal stability despite their intrinsic dynamic behaviour, as evidenced by decomposition temperatures ($T_{d,5\%}$) exceeding 190 °C (Supplementary Figs. 100–102). A three-armed crosslinker synthesized from glycerol (GlyLp₃) was then screened with the same reactive diluents and similar structure–property trends were largely observed (Supplementary Figs. 132–141). Analysis of a material composed of EtLp₁–GlyLp₃ revealed a notably large rubbery plateau extending from roughly 10–160 °C as assessed by dynamic mechanical analysis (Supplementary Fig. 109), which is similar to previous lipoate networks[32]. Materials containing GlyLp₃ generally possessed superior dimensional stability (rubbery plateau of more than or equal to 100 °C) as compared to IsoLp₂-based resins (Supplementary Figs. 103–112), illustrating their large operational window. Overall, a wide range of mechanical properties was accessible from this small pool of lipoates where the UTS, Young's modulus and strain energy density each varied between roughly 1.5 and 2.0 orders of magnitude (Fig. 3f). Many of the compositions are already within the range of 'soft' commercial resins, such as FormLabs' Elastic 50A or Flexible 80A and Photocentrics' Flexible UV160TR (Supplementary Table 6). Our current efforts to further increase mechanical performance are focused on combining the inherent flexibility endowed by their low glass transition temperatures with network reinforcement through non-covalent interactions such as metal coordination strategies[37], stereochemical effects[41] or hydrogen bonding groups[42,43]. The use of acylhydrazines[44], which introduce strong hydrogen bonding interactions, has yielded lipoate networks with UTS near 50 MPa and Young's modulus up to 340 MPa, thus demonstrating that substantial stiffening of the lipoate resin platform could be achievable to compete more broadly with state-of-the-art commercial resins.

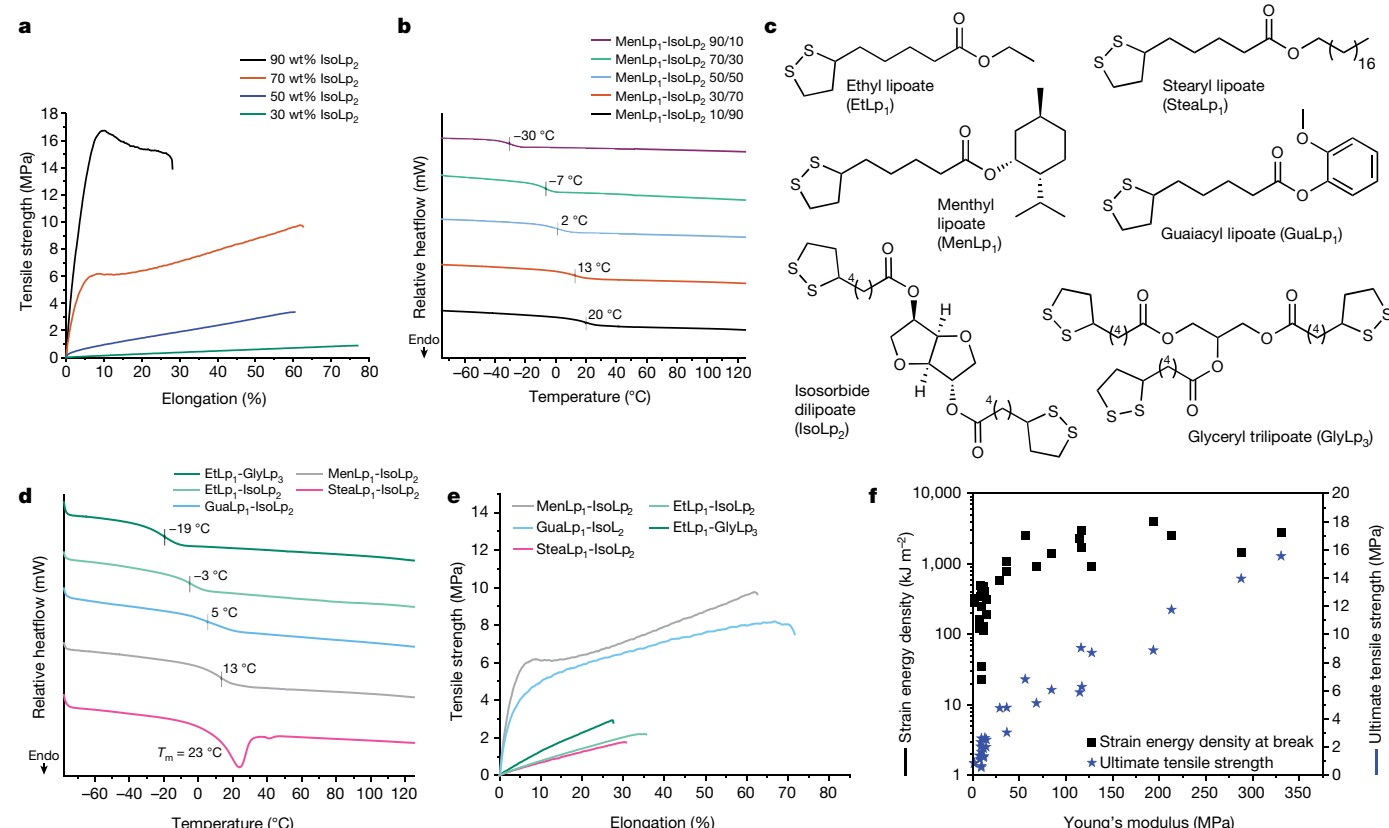

**Fig. 3 | Thermal and mechanical properties of postcured 2D photosets obtained from several renewable lipoate resins. a**, Representative stress versus strain of various MenLp$_1$–IsoLp$_2$ formulations tested at 10 mm min$^{-1}$. **b**, DSC thermograms of the second heating cycle of various MenLp$_1$–IsoLp$_2$ formulations from −80 to 125 °C, 10 °C min$^{-1}$. **c**, Chemical structures of monomeric lipoates (RLp$_1$) and multivalent lipoates. **d**, DSC thermograms of second heating cycle of various lipoate formulations from −80 to 125 °C, 10 °C min$^{-1}$. **e**, Representative stress versus strain of various lipoate formulations (all at 30:70 wt%) tested at 10 mm min$^{-1}$. **f**, Mechanical property scope for surveyed lipoate formulations. Data points are average values ($n$ = 2–5).

Several postcured 2D photosets were catalytically depolymerized, without subsequent optimization from conditions for the 3D-printed MenLp$_1$–IsoLp$_2$, using the same catalyst at higher concentration (10 mol% phosphazene:thiophenol). We observed varying levels of depolymerization efficiency and resin recovery across the range of 2D photosets studied (24–96%, Supplementary Table 2) as assessed by $^1$H NMR spectroscopy. In this unoptimized system, depolymerization of the EtLp$_1$–GlyLp$_3$ was comparable to that of the MenLp$_1$–IsoLp$_2$ 2D photoset. Extension of the study to 3D-printed material led to the recovery of recycled resin in 97% yield with 96% cyclic disulfide content. The ratio of EtLp$_1$–GlyLp$_3$ in the recycled resin (30:70 wt%) was consistent with the initial formulation at 31:69 wt% (Supplementary Fig. 31). We proposed that the lower-than-expected recoveries for the 2D photosets, as compared to the 3D print, could be a result of altered material composition due to overcuring (photo-oxidation) of disulfide species[45,46], although we have not directly observed differences in the infrared spectra[47] between 2D photosets and 3D-printed parts (Supplementary Figs. 78–82). Whereas this phosphazene-catalysed depolymerization was efficient for some formulations, it was not universal and phosphazenes also pose a considerable cytotoxicity hazard[48]. Thus, inspired by the work of Odelius and coworkers in which the ceiling temperature of polylactide was reduced by favourable solvent interactions on the monomer–polymer equilibrium[49], we developed a catalyst-free thermal depolymerization method in which the lipoate networks were simply refluxed in dimethylformamide (DMF) for 2 h. We observed that this thermal depolymerization was effective and generally more consistent for the depolymerization of 2D photosets (yields greater than or equal to 80%) (Supplementary Table 3), and equally applicable to the depolymerization of the

3D-printed EtLp$_1$–GlyLp$_3$ part (91% yield, 96% cyclic disulfide content). Mass spectrometry analysis was also conducted on these depolymerized mixtures but did not offer additional understanding on their composition (Supplementary Figs. 38–46).

The efficient depolymerization and overall high recovery of recycled resin for the cured EtLp$_1$–GlyLp$_3$ materials led us to investigate it more closely in a closed-loop 3D-printing cycle in which we successfully completed two recycle sequences. The EtLp$_1$–GlyLp$_3$ (31:69 wt%) resin was successfully printed and then efficiently depolymerized in high yield (91% for first recycle; 94% for second recycle) using the thermally assisted depolymerization in DMF. The compositions among the pristine and recycled resins were comparable, as determined by $^1$H NMR spectroscopy (Fig. 4a and Supplementary Figs. 33 and 34). The cyclic disulfide content in the recycled resins was determined to be 96% for the first recycle and 94% for the second recycle. The resin composition also remained consistent with a relative molar ratio between EtLp$_1$ and GlyLp$_3$ of 1.32:1, or 32:68 wt% for the first cycle and 1.27:1, or 31:69 wt% for the second cycle, both of which are within error of the pristine resin composition (Supplementary Figs. 33 and 34). SEC analysis of all three resins showed that they had similar molecular weights, although minor amounts of oligomeric species (molecular weight greater than 1,000 g mol$^{-1}$) are observed in the recycled products (Fig. 4b) corroborating the $^1$H NMR spectroscopy data. The ultraviolet-visible light (UV-vis) spectroscopic analysis also showed a slight difference in absorption profile with the appearance of a shoulder around $\lambda$ = 290 nm in the recycled resins, which we also attribute to the small oligomer impurity (Supplementary Fig. 183)[30].

Photorheological analysis of the EtLp$_1$–GlyLp$_3$ resin samples revealed rapid and consistent curing kinetics (Fig. 4c and Supplementary

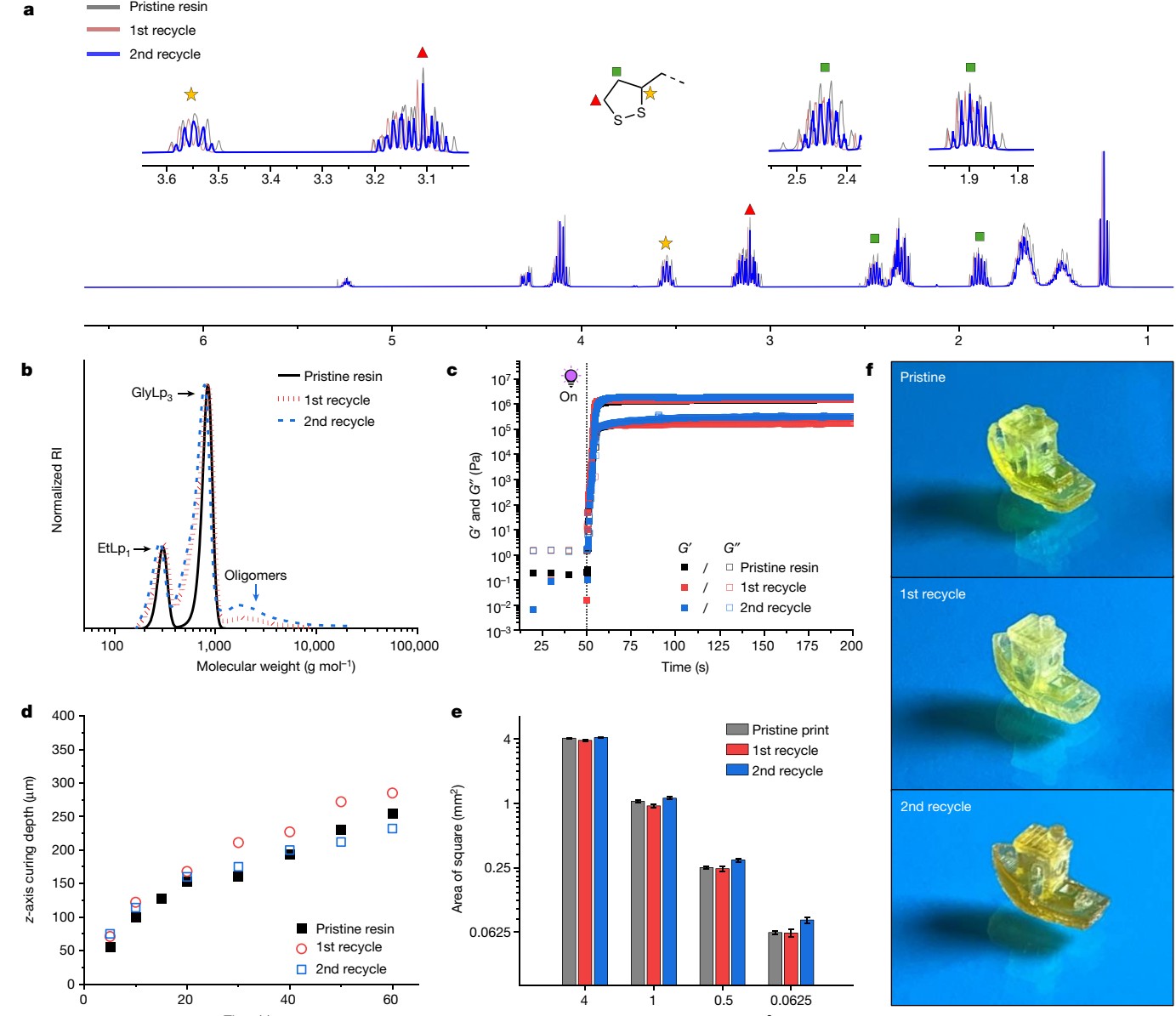

**Fig. 4 | Circular DLP printing of EtLp₁–GlyLp₃ (31:69 wt%) resin. a**, ¹H NMR spectroscopic analysis of pristine resin (300 MHz, 300 K, CDCl₃) compared to recycled resins (400 MHz, 298 K, CDCl₃). **b**, SEC images of pristine resin compared to recycled resins (CHCl₃ + 0.5% v/v NEt₃, against polystyrene standards). **c**, Photorheology at ambient temperature of pristine resin compared to recycled resins under oscillatory shear (0.2 Hz with an amplitude of 25% for 50 s) without irradiation, then the sample was irradiated after 50 s. **d**, Z depth cure screening of pristine compared to recycled resins by irradiating a 2D-square and measuring sample thickness (z axis depth) versus irradiation time (5–60 s).

**e**, x–y printing accuracy for pristine resins compared to recycled resins determined by comparing surface area of squares to theoretical square size (pixel size 30 μm). Theoretical surface area of each square and number of squares (sample size): 4 mm² (n = 3); 1 mm² (n = 4); 0.25 mm² (n = 5); 0.0625 mm² (n = 6). Centre value is average surface area and error bars indicate 1 standard deviation. **f**, 3D-printed parts of '3DBenchy' for pristine resin compared to recycled resins. A photoinitiator was added to recycled resins (1.5 wt% for the first recycle; 2.5 wt% for the second recycle) for photorheology and printing.

Fig. 168). The curing kinetics for the crude recycled resins were slightly slower (Supplementary Figs. 166 and 167), however, this can be rectified by adding photoinitiator after which all resins reached a gel point within 1.4 s. The small decrease in curing rates can probably be attributed to the minor oligomer impurity and its effect on the absorption profile of the recycled resins. Nevertheless, the three cured resins possessed similar plateau moduli, indicating negligible effects on broader bulk material properties from adding photoinitiator to the formulation (Fig. 4c). The z depth cure screening of the same resins also corroborated the constancy among their curing profiles (Fig. 4d), which translated into high-quality x–y printing precision of square arrays with only a slight decrease in accuracy observed for the second recycle resin print

(Fig. 4e and Supplementary Figs. 172–177). Minimal cure-through along the z axis was observed among the three resins (ranging from 123 ± 10 to 129 ± 21%) as assessed from printing an over-hanging bridge (Supplementary Figs. 178 and 179). Together, this enabled the successful printing and reprinting of complex 3D parts with retained features. Slight colour differences were observed, probably due to added photoinitiator. (Fig. 4f and Supplementary Figs. 75–77). To address any ageing and/or successive recycling effects on the resin, we also established a further potentially infinite closed-loop recycling pathway. To this end, hydrolytic depolymerization (NaOH in H₂O/DMF) of the resin was used to regenerate the original resin components, namely alcohols and lipoic acid, (Supplementary Table 4 and Supplementary

Figs. 35–37, 75 and 97–99), which could be easily integrated back into the material cycle to form virgin resin.

These results establish a proof-of-concept advancement in the field of additive manufacturing of photopolymer resins in which circular DLP printing is demonstrated. The use of renewable, sustainable and non-hazardous lipoates simultaneously addresses these limitations of state-of-the-art resins and holds great promise for broader adoption. Lipoate-based resins also offer a health and safety advantage when compared to (meth)acrylate photopolymer resins, which are sensitizers[50,51] that can leach from printed parts as unreacted monomers[52]. Moreover, lipoic acid is already produced at scale and found in commodity products such as health supplements. On account of the chemistry in the resin design, these lipoate-based materials are anticipated to be biodegradable. Whereas hypothetical waste management challenges relating to collection to enable recycling are worthy of discussion, these challenges could be addressed with an evolved collection system that is tailored towards more technical application areas and may find more immediate impact in recycling partially cured waste resin. Current efforts are focused on improving the orthogonality of the network depolymerization, that is, eliminating the presence of oligomeric contaminants, to mitigate the minor differences in resin compositions among recycles at scale.

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

# Methods

## Materials

All compounds, unless otherwise indicated, were purchased from commercial sources and used as received. Lipoic acid was obtained from the sports nutrition brand Myvegan. Spectroscopic analysis of the Myvegan product determined it was similar in composition to a product obtained from a conventional chemical supplier (Acros). MeTHF was distilled before use to remove the radical inhibitor.

## Characterization and processing of materials

NMR spectroscopy experiments were performed at 298 K on a Bruker DPX-400 NMR spectrometer operating at 400 MHz for $^1$H (100.57 MHz for $^{13}$C). $^1$H-NMR spectra are referenced to residual protic solvents (CHCl$_3$ at $\delta$ = 7.26 ppm, DMSO$-d_5$ at $\delta$ = 2.50, CD$_3$DO at $\delta$ = 3.31) and $^{13}$C NMR spectra are referenced to the residual solvent signal (CDCl$_3$ at $\delta$ = 77.16 ppm, $\delta$ = 39.52 for DMSO$-d_6$). The resonance multiplicities are described as s (singlet), d (doublet), t (triplet), q (quartet) or m (multiplet).

High-resolution mass spectrometry was performed by University of Birmingham school of Chemistry on a Waters GCT Premier using electrospray ionization and on Waters Xevo G2-XS using chemical H$^+$ ionization.

Fourier transform infrared spectroscopy was carried out using an Agilent Technologies Cary 630 fourier transform infrared spectrometer. A total of 16 scans from 600 to 4,000 cm$^{-1}$ were taken at a resolution of 2 cm$^{-1}$, and the spectra were corrected for background absorbance.

UV-vis spectroscopy was performed on an Evolution 350 UV-vis spectrophotometer equipped with Xenon Flash Lamp light source and Dual Matched Silicon Photodiodes detector. Quartz cells (170–2,000 nm) from Hellma with two polished sides were used for examining the absorption spectra from 200 to 800 nm in dichloromethane (1 mg ml$^{-1}$) as a solvent. Thermo INSIGHT-2 v.10.0.30319.1 software was used for data acquisition and analysis.

Gel permeation chromatography (GPC) analyses were performed in CHCl$_3$ on an Agilent 1260 Infinity II Multi-Detector GPC–SEC System fitted with a refractive index (RI) detector, ultraviolet (UV, $\lambda$ = 309 nm) and viscometer detectors. The polymers were eluted through an Agilent guard column (PLGel 5 μM, 50 × 7.5 mm) and two Agilent mixed-C columns (PLGel 5 μM, 300 × 7.5 mm) using CHCl$_3$ (buffered with 0.5% NEt$_3$) as the mobile phase (flow rate 1 ml min$^{-1}$, 40 °C). Number average molecular weight ($M_n$), weight average molecular weight ($M_w$) and dispersity ($Đ_M = M_w/M_n$) were determined using Agilent GPC–SEC software (v.A.02.01) against a 15-point calibration curve (peak maxima molecular weight, $M_p$ = 162–3,187,000 g mol$^{-1}$) based on polystyrene standards (Easivial PS-M/H, Agilent).

Differential scanning calorimetry (DSC) thermograms were collected on a STARe system DSC3 with an auto-sampler (Mettler Toledo). Thermograms were obtained in 40 μl aluminium pans from −80 to 125 °C at a heating rate of 10 °C min$^{-1}$ for two heating–cooling cycles unless otherwise specified. The glass transition temperature ($T_g$) was determined by the minimum of the first derivative in the second heating cycle of DSC. Total enthalpy of melting ($\Delta H_m$) was calculated from integration and normalization of all endothermic peaks present.

Thermal gravimetric analysis thermograms were performed using a Q550 Thermogravimetric analyser (TA Instruments). Thermograms were recorded under an N$_2$ atmosphere at a heating rate of 10 °C min$^{-1}$ from 25 to 600 °C. Decomposition temperatures were reported at the 5% weight-loss-temperature ($T_{d,5\%}$)

Dynamic mechanical thermal analysis data were obtained using a Mettler Toledo DMA 1 star system and analysed using the software package STARe v.13.00a (build 6917). Thermal sweeps were conducted using films (length, $L$ × width, $W$ × thickness = 8.60 × 6.20 × 0.60 ± 0.20 mm) cooled to −180 °C and held isothermally for roughly 5 min. Storage and loss moduli, as well as the loss factor (ratio of $E''$ and $E'$, tan $\delta$) were probed as the temperature was swept from −180 to 180 °C, 5 °C min$^{-1}$, 1 Hz.

Uniaxial tensile testing was performed using a Testometric M350-5CT universal mechanical testing instrument fitted with a load cell of 5 kg. Dumbbell shaped samples were cut using a custom ASTM Die D-638 Type 5. Each specimen was clamped into the tensile holders and subjected to an elongation rate of 10 mm min$^{-1}$ until failure. All tensile tests were repeated on at least three individual specimens (unless otherwise indicated), and an average of the data was taken to find the ultimate tensile stress and strain. Strain energy density was calculated from the area under the tensile curves using OriginPro software. The rest of the data were analysed using winTest Analysis software (v.5.0.34) and OriginPro software. Data were analysed using winTest Analysis software (v.5.0.34) and OriginPro software.

Photorheology experiments were used to determine the crosslinking kinetics of the resins as a function of gelation time using an Anton Paar MCR-302 rheometer fitted with a detachable photoillumination system (Exfo OmniCure S1500 UV light source, broadband Hg-lamp, glass plate). Resin samples were sheared between two parallel plates at 0.2 Hz with an amplitude of 25% for 50 s without irradiation. After this time, the light source was switched on and measurements were taken every 0.2 s over the course of 3 min. The intersection point of the storage moduli and loss moduli plots was used to determine the time of gelation of the resin.

Photocuring of 2D-photoset materials was performed by depositing roughly 2 ml of the liquid resin onto a rectangular glass slide (covering the entire slide surface area), followed by curing with an OmniCure S1500 UV light source fitted with a fibre optic cable. Cured films (roughly 0.5 mm thickness) were postcured in an oven at 60 °C for 24 h before assessing their thermomechanical properties.

DLP 3D printing was performed on an unmodified MiiCraft Ultra 125 printer. To assess the $z$ depth curing profile, the resin was deposited on a glass slide and a square shape was irradiated from below the glass slide using a MiiCraft Ultra 125 3D printer as the light source. The excess resin was removed using a tissue and the cure depth was measured using a micrometer. The $z$ depth profiling was performed for each batch of resin to inform the selected print parameters (Fig. 4d and Supplementary Figs. 147 and 148). Printing precision was determined from a 3D-printed custom-made array of square, walls and bridges with various feature sizes (outlined in Fig. 2b and Supplementary Fig. 159) at different layer curing times. The size of the 3D-printed features was visualized using an Alicona G4 InfiniteFocus system and subsequently converted into a 2D height map. The 2D height map images were processed into black and white using ImageJ v.1.53a with the split to binary feature (Fig. 2b and Supplementary Figs. 149, 151, 152, 155, 169 and 170). Finally, the features length and/or area of features were measured within ImageJ and compared to the theoretical feature size (Figs. 2b and 4d and Supplementary Figs. 150, 153, 154, 156–158 and 171–178). Each measurement was performed in triplicate and the error shown was calculated from the standard deviation of the sample set.

## Data availability

All data are available in the paper or the supplementary materials. Source data are provided with this paper.

**Acknowledgements** This project has received funding from the European Research Council (ERC) under the European Union's Horizon 2020 research and innovation programme (Grant agreement nod. 681559 and 963898) (J.C.W., C.J.S., A.P.D. and T.O.M.). This project has also received funding from the European Union's Horizon 2020 research and innovation programme under the Marie Skłodowska-Curie grant agreement no. 101030883 (V.C.). This project has received funding from UK Research and Innovation under the UK government's Horizon Europe funding guarantee under the grant agreement number EP/X022838/1 (T.O.M.).

**Author contributions** J.C.W. and A.P.D. conceived the work. All authors designed the experiments. C.J.S., T.O.M., V.C., M.A.A., A.B. and J.C.W. performed and analysed experiments. J.C.W. and A.P.D. directed the research. J.C.W. and A.P.D. prepared the manuscript and all authors contributed to manuscript revisions.

**Competing interests** J.C.W., A.P.D., C.J.S. and T.O.M. are inventors on a provisional patent application, submitted by the University of Birmingham, relating to this work.

**Additional information**
**Correspondence and requests for materials** should be addressed to Joshua C. Worch or Andrew P. Dove.
