## [Peer Review File · Nature]

Manuscript Title: A renewably sourced, circular photopolymer resin for additive manufacturing

Reviewer Comments & Author Rebuttals

Reviewer Reports on the Initial Version:

Referees' comments:

Referee #1 (Remarks to the Author):

In this manuscript, Dove, Worch, and co-workers build upon their strong expertise in circular polymers and additive manufacturing with a contribution that describes 3D printing of a photopolymer resin comprising lipoate functionalized monomers derived from renewable (biobased) sources. The manuscript covers an important topic of polymer circularity in the context of an emergent technology (additive manufacturing), and showcases what is likely the most circular photopolymer resin to-date. Although scientifically sound (i.e., conclusions are validated by experiments, proper controls are provided, etc.), the work is not viewed by this reviewer as being novel enough or transformative enough to merit publication in Nature. However, this work does represent an important advance to closed-loop recycling in 3D printing and the authors do provide an excellent composition-thermomechanical relationship survey for monomers derived from various renewably sourced alcohols. Thus, this work will garner interest among the associated polymer communities and as such it could represent an excellent contribution to Nature Communications as a possible alternative. Below are more specifics associated with perceived novelty and impact along with subsequent comments/questions for the authors to consider.

To the point of novelty, the work directly builds on prior art. For example, the utility of disulfides, and more specifically lipoates, in the fabrication of dynamic and reversible polymer networks has previously been reported in the literature, as recognized by the authors (Refs 28-33). Of particular relevance is the work by Hawker, Read de Alaniz, Bates, and co-workers where 3D printing of lipoate-functionalized monomers was accomplished (Ref 29), albeit this work represented a combination of thiol-ene addition and disulfide formation so samples could only be healed as opposed to recycled as demonstrated here. Furthermore, and possibly more relevant, was work by Tian, Qu, and coworkers (Ref 30) along with Feringa, Zhang and coworkers (Ref 33) where circularity of disulfide polymers prepared from lipoate derivatives was showcased, the main distinction noted herein being the ability to 3D print them and specifically appending them to common biorenewable diols that have been employed by the present authors in a number of other publications, as well as numerous other cases in synthesizing, for example, polyesters with the common objective of improving plastic circularity.

To the point of impact, the authors clearly showcase the ability to 1) perform high resolution and fast light-based 3D printing of resins comprised of biorenewable monomers and 2) depolymerize the 3D printed structures and reprint from that same resin, which together is meritorious. However, to the

point of extending and applying this platform it was less clear given the potential thermal stability of printed parts and the overall mechanical performance. While the authors do demonstrate the ability to tune the thermal properties (i.e., glass transition temperature) and mechanical properties (i.e., ultimate tensile strength, stiffness, and strain energy density), the ranges provided are not competitive with commercial thermoset materials, for which these (or derivatives of these) would presumably replace. For example, extensibility beyond 100% for soft, elastomeric materials has been achieved with 3D printed acrylics, along with ultimate tensile strengths in excess of 60 MPa. Additionally, the authors do not note how one would potentially build on this nice suite of resin building blocks to further enhance the materials performance apart from simply stating that “different ratios of crosslinker to diluent” could be explored (page 6). Given the intrinsic flexibility and dynamic nature of disulfide bonds, it seems challenging to overcome these issues, which is something the authors should more directly address to indicate how further work in this class of polymers could lead to advanced materials.

Comments/questions:

- 1) Minor point – in the introduction the authors state that “...most 3D-printed parts are non-recyclable (i.e., not depolymerizable)...”, but it would be more appropriate to say instead (e.g., not depolymerizable) as depolymerization is one example of how recycling can be achieved, while i.e., suggests that it is the only (or predominant) method, which is less accurate.
- 2) The first example of the reversible photopolymerization of disulfides was reported back in 1954 by Calvin and coworkers (DOI: 10.1021/ja01646a029). The authors are encouraged to include this seminal reference.
- 3) Minor point – typo in Reference 33 “Polymer Reviews” should be removed from the author list.
- 4) Minor point – typo on page 3 with reference to “Fig. 1A”, should be “Fig. 2A”.
- 5) On page 3 the authors note that stability is influenced by the resin composition (different monomer ratios). Shelf stability is a major industrial consideration and thus having quantification of this attribute (such as rheology over time) and discussion on why certain mixtures are more stable than others, would help guide further advancements.
- 6) A side-by-side comparison of resolution for pristine and recycled resin would be helpful in comparing the results. Currently all of the quantitative resolution data is present in the SI file – apart from Figure 2C, which provides only a small snapshot of the data. It would additionally be helpful to provide more quantitative details of feature resolution in the main text – what are the smallest achievable features and how many pixels on the projection do those features represent? What is the percent overcure observed and how does that compare to commercial resin?
- 7) The authors provide values for “toughness”, but do not describe how those values were obtained. They are likely area under the stress-strain curve, in which case it should be referred to as “strain energy density” not “toughness” as it is not a true measure of toughness, although this terminology is commonly misused in the literature.
- 8) The authors noted that “cure-through was inconsistent among samples without photoinitiator present”, but it wasn’t entirely clear why that was the case. It would be helpful if the authors could clarify why TPOL (photoinitiator) addressed this issue and further clarify for what sample preps it was used. For example, descriptions of tensile testing immediately follow the mention of TPOL, but were the samples mechanically tested 3D printed or molded and how does cure-through factor in here?

9) On page 7 the authors note that the gel points for these resins were observed at between 4-9 seconds (pristine vs. recycled), however layer exposure times for the prints were 35 seconds. In acrylic resins the layer cure time is usually closer to the gel point observed by photorheology. To this point, it would be helpful for the authors to explicitly state what light intensities/wavelengths were used for both processes along with the layer thicknesses used.

10) It would be helpful if the authors provided more thermal characterization of these materials, such as thermogravimetric analysis (TGA) and dynamic mechanical analysis (DMA) to more directly probe their operational windows.

Referee #2 (Remarks to the Author):

This manuscript describes a new photopolymer resin platform that allows high resolution printing and can be depolymerized and reprinted. This work is very important considering the sustainability issues associated with the use of photocrosslinked resins in additive manufacturing. The work is based on the use of photoreactive cyclic disulfides. Points that should be considered by the authors:

As a proof of concept, this work opens very interesting avenues for the development of new resins that can be reused. The authors have tried to link their work to a green process namely the use of compounds from renewable resources to obtain their monomers. However, the authors used toxic solvents in the production of the monomers and phosphazene in the depolymerization process.

Phosphazene is very toxic even in very low concentrations. See reference as an example (<https://pubmed.ncbi.nlm.nih.gov/28835095/>). How the use of these reactants could negatively impact the sustainability of the process? The authors should discuss this.

The products of depolymerization should be characterized by mass spectroscopy to better understand the compositional differences. NMR is not sufficient to understand the structures formed.

The mechanical data of the materials should be compared with another classical acrylate used for SLA. It would be important to understand the E' profile (by DMTA) of these materials knowing that some formulations were made with 70% crosslinker.

It is mentioned that " SEC analysis of the recycled resin showed that it had a similar molecular weight profile to the original resin, although a small amount of polymeric product was observed, which indicates that the depolymerization was not 100% effective (Fig. 2G)." Did the authors try to extend the depolymerization time (or change other conditions?) to eliminate polymer residues? A Soxhlet extraction should be performed to evaluate the gel content of the crosslinked and depolymerized material. The products resulting from the extraction should be characterized.

Referee #3 (Remarks to the Author):

This authors describe the synthesis of a platform of new bio-based photopolymer resins for additive manufacturing, followed by depolymerization of printed objects and subsequent re-printing. The manuscript is well-written, concise and of high quality. I agree with the authors that the closed-loop recycling of stereolithographic printed parts is a scientific challenge, and while several research teams

attempted to reach that goal, most ended up with an open-loop process, since addition of extra resin components was needed for re-printing. The authors suggest their approach is closed-loop. However, to ensure reliable re-printing, they had to add extra photoinitiator to the recycled resin. Therefore, in my opinion, it is not so different with respect to what they have defined as "open-loop".

The following points need attention:

- Page 4 line 79-81: Partial gelation was avoided by reacting both alcohols simultaneously with lipoic acid. What was the reason for the gelation? And why was it circumvented by reacting with both alcohols at the same time? This clearly needs more explanation in the manuscript.
- Page 4 line 91: The pristine resin was formulated without a photoinitiator. Could the authors elaborate on the actual stability of their resin at ambient conditions?
- Page 5 line 116: The depolymerization was not 100% effective. It seems the researchers were able to re-print once, which is already an excellent achievement. However, to what extent is it possible to recycle multiple times (over multiple cycles)?
- Page 7 line 190: The authors mention a small decrease in curing rate. However, it takes twice as long to reach the gel point for the recycled resin. That seems quite a difference. Please discuss.
- Page 7 line 192: In the introduction, the authors refer to open-loop processes that need addition of extra resin components for re-printing (and re-curing). They claim their process is, in contrast to others, closed-loop (Figure 1C). However, to ensure reliable re-printing, they add a small quantity of photoinitiator to the recycled resin, while this was not needed for the first cycle. From that viewpoint, to what extent is this a real closed-loop process?

Author Rebuttals to Initial Comments:

Lipoic paper

Referee #1 (Remarks to the Author):

In this manuscript, Dove, Worch, and co-workers build upon their strong expertise in circular polymers and additive manufacturing with a contribution that describes 3D printing of a photopolymer resin comprising lipoate functionalized monomers derived from renewable (biobased) sources. The manuscript covers an important topic of polymer circularity in the context of an emergent technology (additive manufacturing), and showcases what is likely the most circular photopolymer resin to-date. Although scientifically sound (i.e., conclusions are validated by experiments, proper controls are provided, etc.), the work is not viewed by this reviewer as being novel enough or transformative enough to merit publication in Nature. However, this work does represent an important advance to closed-loop recycling in 3D printing and the authors do provide an excellent composition-thermomechanical relationship survey for monomers derived from various renewably sourced alcohols. Thus, this work will garner interest among the associated polymer communities and as such it could represent an excellent contribution to Nature Communications as a possible alternative. Below are more specifics associated with perceived novelty and impact along with subsequent comments/questions for the authors to consider.

To the point of novelty, the work directly builds on prior art. For example, the utility of disulfides, and more specifically lipoates, in the fabrication of dynamic and reversible polymer networks has previously been reported in the literature, as recognized by the authors (Refs 28-33). Of particular relevance is the work by Hawker, Read de Alaniz, Bates, and co-workers where 3D printing of lipoate-functionalized monomers was accomplished (Ref 29), albeit this work represented a combination of thiol-ene addition and disulfide formation so samples could only be healed as opposed to recycled as demonstrated here. Furthermore, and possibly more relevant, was work by Tian, Qu, and coworkers (Ref 30) along with Feringa, Zhang and coworkers (Ref 33) where circularity of disulfide polymers prepared from lipoate derivatives was showcased, the main distinction noted herein being the ability to 3D print them and specifically appending them to common biorenewable diols that have been employed by the present authors in a number of other publications, as well as numerous other cases in synthesizing, for example, polyesters with the common objective of improving plastic circularity.

We agree with the reviewer's point that the dynamic nature of cyclic disulfides, particularly lipoates, is well established in the field. Historic reports from the 1950s established these principles for linear polymers: "Disulfide Polymers of DL- α -Lipoic Acid" *J. Am. Chem. Soc.* 1956, 78, 23, 6148–6149; "The preparation and relative reactivities of many-membered cyclic disulfides" *J. Org. Chem.* 1950, 15, 4, 865–868; "Kinetics of the Thiol-Disulfide Exchange", *J. Am. Chem. Soc.* 1957, 79, 4, 833–838; and the report mentioned by the reviewer in comment 2: "The Chemistry of 1,2-Dithiolane (Trimethylene Disulfide) as a Model for the Primary Quantum Conversion Act in Photosynthesis 1a" *J. Am. Chem. Soc.* 1954, 76, 4348-4367. As may be expected, dynamic network formation (and depolymerization) has proven somewhat more challenging. While some recent works have studied dynamic disulfide networks, in our manuscript we advance these approaches in a unique way to furnish, what the reviewer describes as, "the most circular photopolymer resin to date". We note that this is also composed of 100% renewable building blocks. Furthermore, we applied this technology to DLP which currently lacks any circular solutions to enable 3D re-printing of a photocurable resin without significant reformulation efforts or the addition of (reactive)

diluents. This is a substantial leap from the state-of-the-art in recyclable photoresins (Ref 23-24) which are both clearly open-loop processes. In these cases, the only way to recycle the resin is to add significantly more monomer in each cycle. We direct Reviewer 1 to our response to Reviewer 3's final comment, which provides an in-depth analysis of the advantages of our method compared to these prior reports.

We specifically targeted lipoate technology due to 1) its ability to undergo radical (photoinitiated) ROP and 2) the near equilibrium nature of its polymers (depolymerizability). However, one of the reports (Ref 29, now Ref 30; *Adv. Funct. Mater.* 2022, 32, 2200883), which is noted by Reviewer 1, showed that curing of a lipoate-only resin was significantly slower than a mixed acrylate-lipoate system and the implementation of this dual acrylate-lipoate system was "...crucial for efficient DLP printing" (Ref 29, now Ref 30). Moreover, a historical report mentions similar issues with the photopolymerization of cyclic disulfides due to "low primary quantum yield." (*Trans. Faraday Soc.*, 1957, **53**, 813-820). As such, the successful 3D printing of an all-lipoate resin using DLP is an unprecedented invention in the field establishing a distinctly new curing technology for 3D printing that avoids acrylates and/or other conventional resin components.

We agree with the reviewer that Ref 33 (now Ref 36) by Feringa and co-workers is an important work to discuss. The work of "Tian, Qu, and coworkers (Ref 30, now Ref 33)" is a review article so not relevant to this discussion. Perhaps the Reviewer meant Ref 31 (now Ref 34) by Bates and co-workers? While these are highly relevant, we disagree with the modest characterization of our advance in relation to these prior reports. The study by Feringa and co-workers depolymerised thin 2D films of lipoate-based networks and demonstrated only partial recovery of the dithiolane species. Photopolymerization of these 'recovered' species, i.e. recuring, was not disclosed. Furthermore, translating this formulation to 3D structures seemed unfeasible, even for the pristine formulation, let alone using the recovered (recycled) components (Ref 33, now Ref 36; *Chem. Commun.*, 2021, 57, 9838-9841). Bates and co-workers showed that a lipoate-only network (with relatively low crosslinking density) can be thermally depolymerised to recover 30-40% of the dithiolane functionality (Ref 31, now Ref 34; *J. Am. Chem. Soc.* 2021, 143, 26, 9866-9871). Although this material could be re-cured into a thin film with UV irradiation, the recycled film displayed a 200-fold decrease in modulus (*initial* – 200 kPa, *recycled* – 1 kPa). While linear polyliipoates are often reported to depolymerize with moderate to high conversion, networks have proved more challenging as illustrated by Ref 34 & 36.

In contrast to these preceding lipoate network depolymerisations, we now report the near quantitative depolymerisation of a 100% renewable 3D-printed part (> 95% cyclic disulfide, >90% yield) using mild conditions without catalyst (140 °C in DMF, 2 h) – this is a newly developed procedure that avoids toxic organobase catalysts (the hazards of phosphazenes were pointed out by Reviewer 2). This recovered resin can be re-printed, depolymerized and then re-printed again (new data disclosed here in Fig 4 and accompanying discussion) while maintaining consistent curing profiles and high-precision printed parts. These advances not only provide the most circular photopolymer to date, but its high efficiency in terms of curing, depolymerisation, and robust formulation render it suitable for high-fidelity, sustainable 3D printing. The state-of-the-art accounts fail to satisfy several of these critical criteria.

To the point of impact, the authors clearly showcase the ability to 1) perform high resolution and fast light-based 3D printing of resins comprised of biorenewable monomers and 2) depolymerize the 3D printed structures and reprint from that same resin, which together is meritorious. However, to the point of extending and applying this platform it was less clear given the potential thermal stability of printed parts

and the overall mechanical performance. While the authors do demonstrate the ability to tune the thermal properties (i.e., glass transition temperature) and mechanical properties (i.e., ultimate tensile strength, stiffness, and strain energy density), the ranges provided are not competitive with commercial thermoset materials, for which these (or derivatives of these) would presumably replace. For example, extensibility beyond 100% for soft, elastomeric materials has been achieved with 3D printed acrylics, along with ultimate tensile strengths in excess of 60 MPa.

Additionally, the authors do not note how one would potentially build on this nice suite of resin building blocks to further enhance the materials performance apart from simply stating that “different ratios of crosslinker to diluent” could be explored (page 6). Given the intrinsic flexibility and dynamic nature of disulfide bonds, it seems challenging to overcome these issues, which is something the authors should more directly address to indicate how further work in this class of polymers could lead to advanced materials.

We agree that a thorough analysis of the thermal stability of printed structures is important. We conducted TGA to assess thermal degradation and DMA temperature sweeps to verify the dimensional integrity of the prints. The lipoate networks possess adequate thermal stability, especially the flagship formulations EtLp₁:GlyLp₃ ($T_{d,5\%} = 196$ °C) and MenLp₁-IsoLp₂ ($T_{d,5\%} = 221$ °C). The DMA thermogram of EtLp₁:GlyLp₃ revealed a large rubbery plateau extending from 10–160 °C. Resins containing GlyLp₃ generally possessed wider rubbery plateaus before failure, as compared to IsoLp₂-based resins, with all showing good dimensional stability up to at least 100 °C. TGA and DMA data for all material formulations are now included (Figure S100-112) with appropriate discussion in the text. The text was amended:

Page 6, line 33: “All lipoate networks possessed adequate bulk thermal stability, despite their dynamic behavior, as evidenced by decomposition temperatures ($T_{d,5\%}$) exceeding 190 °C (Fig. S100-102).”

Page 6, line 37: “Analysis of a material composed of EtLp₁:GlyLp₃ revealed a notably large rubbery plateau extending from *ca.* 10–160 °C as assessed by dynamic mechanical analysis (DMA) (Fig. S109). Materials containing GlyLp₃ generally possessed superior dimensional stability (rubbery plateau ≥ 100 °C) as compared to IsoLp₂-based resins (Fig. S103-112), illustrating their large operational window.”

The purpose of this study is to demonstrate proof-of-concept for the resin technology; thus, in-depth optimization was deprioritized at this stage. Using just 6 resin components, we still achieve a wide range of properties, e.g. both UTS and Young’s modulus can be tuned by over an order of magnitude. However, it is not unreasonable to expect that formulation and/or compositional development could propel these materials into a more competitive stance alongside stronger commercial resins (strategies are elaborated later). We collated common tensile property data for commercial resins (Table S6) and contextualized our lipoate resin platform against these materials to provide greater transparency on the present mechanical challenges. The commercial resins we assessed possess a large distribution of properties, which is unsurprising considering their markedly different compositions: UTS = 3.2–72 MPa, Young’s modulus = 8.0–9300 MPa, Elongation at break = 1.2–400%. Our reported resin platform occupies a property space that is more comparable to softer resins, although *many metrics fit within the range of commercial resins*: UTS range = 0.7–16 MPa, Young’s modulus = 1.4–330 MPa, Elongation at break = 9–70 %)

We have provided additional discussion and reference to the new SI table in the manuscript:

Page 6, line 43: “Many of the compositions are already within the range of “soft” commercial resins, such as FormLabs’ Elastic 50A™ or Flexible 80A™ and Photocentrics’ Flexible UV160TR™ (Table S6).”

It is important to acknowledge that the privileged property scope of state-of-the-art commercial resins has been achieved through a decades-long, global research effort into acrylate and epoxy based-photocuring resins. Thus, we argue that it is an unreasonable expectation to match all these features within a single research project that is centred on demonstrating a new circular 3D printing resin technology. However, lessons learned from acrylate or epoxide resin technology development highlight many ways to improve the mechanical performance, namely by changing resin functionality, incorporating hydrogen bonding motifs, and/or using additives. There is even literature precedent for improving the mechanical performance of lipoate networks. A few examples from Feringa and coworkers report using lipoamides, acylhydrazines (Ref 42, DOI: 10.1126/sciadv.abk3286), thiocarbamates and/or metal interactions (Ref 47, DOI: 10.1002/anie.201913893) to yield tough networks with UTS up to 50 MPa and elongations at break around 200-300%. Another key modification would be the incorporation of inorganic fillers or other composite materials, which is common in high performance printing resins and offer improvements in strength and Young’s modulus (<https://doi.org/10.1016/j.compositesb.2016.11.034>). A distinct advantage of our depolymerizable network would be the streamlined process recovery of any non-reactive fillers incorporated in our resin system. The text has been updated:

Page 6, line 45 – “Our current efforts to further increase mechanical performance are focused on combining the inherent flexibility endowed by their low T_g s with metal coordination strategies(42) or hydrogen bonding groups such as acylhydrazines(47), the latter of which has yielded lipoate networks with UTSs near 50 MPa. Thus, the property space of the lipoate resin platform could be straightforwardly augmented to compete more broadly with state-of-the-art commercial resins.”

Comments/questions:

1) Minor point – in the introduction the authors state that “...most 3D-printed parts are non-recyclable (i.e., not depolymerizable)...”, but it would be more appropriate to say instead (e.g., not depolymerizable) as depolymerization is one example of how recycling can be achieved, while i.e., suggests that it is the only (or predominant) method, which is less accurate.

This is a good clarification to include. The text has been updated to reflect this (page 2 line 16): “e.g. not depolymerizable”.

2) The first example of the reversible photopolymerization of disulfides was reported back in 1954 by Calvin and coworkers (DOI: 10.1021/ja01646a029). The authors are encouraged to include this seminal reference.

We would like to thank the reviewer for pointing out this historical study on dithiolanes, which is now cited on page 2 line 36 (Ref 32): “On account of their ready polymerization by radical-mediated methods (30-32)”

3) Minor point – typo in Reference 33 “Polymer Reviews” should be removed from the author list.

This is now Ref 36 and has been updated.

4) Minor point – typo on page 3 with reference to “Fig. 1A”, should be “Fig. 2A”.

Page 4 line 1 has been changed from (Fig. 1A) to (Fig. 2A)

5) On page 3 the authors note that stability is influenced by the resin composition (different monomer ratios). Shelf stability is a major industrial consideration and thus having quantification of this attribute (such as rheology over time) and discussion on why certain mixtures are more stable than others, would help guide further advancements.

This is great point, and we obtained quantitative stability data on the EtLp₁:GlyLp₃ lipoate resin using photorheology. The stability was analyzed over 35 days for the as-synthesized (or pristine) resin, which was kept at ambient temperature (21-23 °C) and protected from light for the duration of the study (Fig. S165). The resins’ loss modulus remained greater than the storage modulus until day 25, with only minor fluctuations observed, and showed a gel point after irradiation. When assessed on day 35, the storage modulus was found to be greater than the loss modulus (indicating gelation) before irradiation. We are still probing the surprising stability enhancement of resins, especially as several lipoates are mixed or formulated, however this feature seems to be general from anecdotal observations. We expect that this shelf life can be extended through addition of radical inhibitors as is common in most resins. We also note that the partially cured resin can be returned to an uncured state through depolymerisation. The text was amended:

Page 3 line 17, “...we observed that the mixture was comparatively more stable than either component in isolation (Fig. 2A), possibly a consequence each mixed component serving as a diluent to the other, despite also possessing cyclic disulfide motifs.

Page 4 line 10, “The formulated resin could be stored on the bench-top when protected from light for several weeks while maintaining a consistent viscosity and rapid photocurability (Fig. S165). It is important to note that aging effects can be neutralized by depolymerizing back to monomer components.”

We also investigated a hydrolytic depolymerization (NaOH in H₂O/DMF mixture) method to return any deteriorated resin to starting materials (lipoic acid and alcohols). We include several cursory experiments related to this aim to establish feasibility (Fig S35-37 - NMR; Fig. S75-77 – FTIR; Fig. S97-99 – SEC Table S4 – summary of experiments). We have added this discussion to the last section when discussing print recycling:

Page 8, line 8: “To address any aging and/or successive recycling effects on the resin, we also established another “looped” recycling pathway. A hydrolytic depolymerization (NaOH in H₂O/DMF) of the resin was employed to regenerate the original resin components, namely alcohols and lipoic acid, (Table S4, Fig. S35-37, S75, S97-99) which could be easily integrated back into the material cycle to form virgin resin”

6) A side-by-side comparison of resolution for pristine and recycled resin would be helpful in comparing the results. Currently all of the quantitative resolution data is present in the SI file – apart from Figure 2C, which provides only a small snapshot of the data.

Fig. 4 panel E now provides a side-by-side-by-side (pristine, 1st recycle, 2nd recycle) comparison for square array print resolution. The supporting information contains additional resolution data for wall thickness and bridge thickness, which are plotted individually and side-by-side-by-side (Fig. S171-178)

It would additionally be helpful to provide more quantitative details of feature resolution in the main text – what are the smallest achievable features and how many pixels on the projection do those features represent? What is the percent overcure observed and how does that compare to commercial resin?

These are helpful suggestions. The commercial printer has a moderate x-y resolution limit and, hence, a pixel size of 30 μm . The printer slicing software will round to the nearest pixel and use antialiasing when not an integer to achieve a size closest to the desired dimensions. The smallest feature size that we attempted was a 50 μm wall (~ 2 pixel width) but this collapsed in most printing attempts and usually measured larger than the theoretical size when it was achieved. The smallest printed feature that could be *reliably reproduced* was 100 μm (~ 3 -4 pixels) in dimensions, for both a wall and square. We have now provided the pixel size in the Figure 2C caption for the reader. We have also incorporated a line discussing the smallest feature and the pixel width estimate in the text:

Page 4 line 27 “The smallest feature we could reproducibly print was a 100 μm wall (~ 3 pixels wide) at 25s per 50 μm , highlighting the impressive x-y resolution of our resin platform on an off-the-shelf commercial 3D printer.”

Commercial resins are highly formulated trade secrets containing optical absorbers (OA) and/or photoinhibitors to prevent cure-through, unlike our reported system. Nonetheless, we agree that it is important to provide a like-for-like comparison of our lipoate-based resin to the prevailing acrylic resin technology. A similar bridge print was performed on a thiol-ene acrylate resin not containing OA by Page and coworkers (Ref 39; DOI: 10.1002/adma.202104906). The cure-through for a 1000 μm bridge was ($185\% \pm 51$) and a 250 μm bridge was ($308\% \pm 124$) as opposed to our system, also without any OA, at 20s curing ($123\% \pm 10\%$) for a 500 μm bridge. This demonstrates a superior resistance to cure-through in our resin platform when compared to a typical acrylic resin that does not contain OA. We have added some additional context and referenced the cure-through percentage in the text:

Page 4 line 31: “...where only modest cure-through ($113\% \pm 7\%$ at 20 s irradiation) was observed for an over-hanging bridge (Fig. 2B). Commercial acrylic resins routinely incorporate opaquing agents to inhibit significant cure-through which can distort z-resolution as evidenced by poor z-resolution (overcure ≥ 200 -300%) when an acrylic is deliberately cured without an added opaquing agent(45). It is likely that the z-axis resolution for the lipoate-based resins could be further optimized with the addition of an opaquing agent.”

7) The authors provide values for “toughness”, but do not describe how those values were obtained. The are likely area under the stress-strain curve, in which case it should be referred to as “strain energy density” not “toughness” as it is not a true measure of toughness, although this terminology is commonly misused in the literature.

The term “toughness” has been replaced with “strain energy density” on page 6 line 43 and in the y-axis title of the plot in Fig. 3F. A description of how strain energy was calculated is also provided in the methods section of the SI:

Page 3 of SI: “Strain energy density was calculated from the area under the tensile curves using OriginPro® software.”

8) The authors noted that “cure-through was inconsistent among samples without photoinitiator present”, but it wasn’t entirely clear why that was the case. It would be helpful if the authors could clarify why TPOL (photoinitiator) addressed this issue and further clarify for what sample preps it was used. For example, descriptions of tensile testing immediately follow the mention of TPOL, but were the samples mechanically tested 3D printed or molded and how does cure-through factor in here?

The 2D photosets were approximately 0.5 mm in thickness, which is an order of magnitude thicker than the curing depth of the printed layers. At these large thicknesses for the 2D films, we anecdotally observed incomplete gelation without photoinitiator present, so TPOL was added to ensure that these 2D samples were consistently cured from top-bottom (i.e. all depths). The discussion on 2D photosets beginning on page 6 indicated that TPOL was added to produce these materials, although we have clarified this language in the text to avoid confusion with “cure-through”, which is a 3D printing term:

Page 6 line 15: “The photoinitiator was incorporated to provide consistent and spatially even gelation of the resins at this thickness.”

The addition of *photoinitiator is not necessary to print pristine resins*, as already highlighted in the text on page 4 line 22: “The MenLp₁:IsoLp₂ (30:70 wt.%) resin was formulated without the need for additives, *i.e.*, without a photoinitiator,..” Furthermore, this is re-iterated by discussion on page 7 line 12: “We hypothesized that the lower-than-expected recoveries for the 2D photosets, as compared to the 3D print, could be a result of altered material composition due to overcuring (photo-oxidation) of disulfide species,(48, 49) although we have not directly observed differences in the IR spectra(50) between 2D-photosets and 3D-printed parts (Fig. S78-82).”

The Fig. 3 caption expresses that the mechanical data was obtained from 2D photosets “**Fig. 3. Thermal and mechanical properties of post-cured 2D-photosets...**” which is referenced in relation to the tensile testing discussion that the Reviewer points out. The screening of 2D photosets was a strategic choice to enable more rapid exploration of the structure-property space, which is already mentioned in the text (page 6 line 13-15: “We created 2D-photosets...to enable rapid screening of materials” However, Figure S164 shows the tensile curves for EtLp₁:GlyLp₃ 3D prints (pristine and recycled), both of which are slightly weaker but still comparable to data for the EtLp₁:GlyLp₃ 2D photoset (Fig. 3E, Fig. S144). The text was amended to reflect:

Page 6, line 17: “Importantly, these 2D-photosets are suitable surrogates for approximating 3D-printed parts since the printed structures exhibited similar mechanical profiles, despite being slightly weaker overall. (Fig. S144, S164)

9) On page 7 the authors note that the gel points for these resins were observed at between 4-9 seconds (pristine vs. recycled), however layer exposure times for the prints were 35 seconds. In acrylic resins the layer cure time is usually closer to the gel point observed by photorheology. To this point, it would be helpful for the authors to explicitly state what light intensities/wavelengths were used for both processes along with the layer thicknesses used.

For the photorheology experiments, a layer thickness of 50 μm was used. During 3D printing the z-layer height was set to 50 μm. The layer thickness for the photorheology has been included in the SI materials

and methods section. Irradiance measurements of wavelengths spanning 200 – 800 nm of all the light sources used in this study were previously included in the SI (Figure S184).

10) It would be helpful if the authors provided more thermal characterization of these materials, such as thermogravimetric analysis (TGA) and dynamic mechanical analysis (DMA) to more directly probe their operational windows.

This data has been added and our prior response is shown again here for clarity:

We agree that a thorough analysis of the thermal stability of printed structures is important. We conducted TGA to assess thermal degradation and DMA temperature sweeps to verify the dimensional integrity of the prints. The lipoate networks possess adequate thermal stability especially the flagship formulations EtLp₁:GlyLp₃ ($T_{d, 5\%} = 196$ °C); MenLp₁-IsoLp₂ ($T_{d, 5\%} = 221$ °C). The DMA thermogram of EtLp₁:GlyLp₃ revealed a large rubbery plateau extending from 10–160 °C. Resins containing GlyLp₃ generally possessed wider rubbery plateaus before failure, as compared to IsoLp₂-based resins, with all showing good dimensional stability up to at least 100 °C. TGA and DMA data for all material formulations are now included (Figure S100-112) with appropriate discussion in the text. The text was amended:

Page 6, line 33: “All lipoate networks possessed adequate bulk thermal stability, despite their dynamic behavior, as evidenced by decomposition temperatures ($T_{d, 5\%}$) exceeding 190 °C (Fig. S100-102).”

Page 6, line 37: “Analysis of a material composed of EtLp₁:GlyLp₃ revealed a notably large rubbery plateau extending from *ca.*10–160 °C as assessed by dynamic mechanical analysis (DMA) (Fig. S109). Materials containing GlyLp₃ generally possessed superior dimensional stability (rubbery plateau ≥ 100 °C) as compared to IsoLp₂-based resins (Fig. S103-112), illustrating their large operational window.”

Referee #2 (Remarks to the Author):

This manuscript describes a new photopolymer resin platform that allows high resolution printing and can be depolymerized and reprinted. This work is very important considering the sustainability issues associated with the use of photocrosslinked resins in additive manufacturing. The work is based on the use of photoreactive cyclic disulfides. Points that should be considered by the authors:

As a proof of concept, this work opens very interesting avenues for the development of new resins that can be reused. The authors have tried to link their work to a green process namely the use of compounds from renewable resources to obtain their monomers. However, the authors used toxic solvents in the production of the monomers and phosphazene in the depolymerization process. Phosphazene is very toxic even in very low concentrations. See reference as an example (<https://pubmed.ncbi.nlm.nih.gov/28835095/>). How the use of these reactants could negatively impact the sustainability of the process? The authors should discuss this.

These are excellent broader points regarding the overall sustainability of the approach, and we address them by offering potential alternatives and modifying some processes. The use of Steglich coupling to synthesize the lipoate resins was chosen as a simple and reliable procedure to produce the necessary quantities required for 3D printing. We do acknowledge that it has poor performance metrics from a green chemistry standpoint, though it remains an important coupling method for commercial production of fine chemicals and pharmaceuticals (with esters DOI: 10.1021/acs.chemrev.0c00709 and amides DOI:

10.1039/B701677H), which are more analogous in scale to photopolymer resins rather than bulk commodity chemicals. Still, there have been attempts to reduce the impact by substituting solvents and reagents. A report by Sneddon and co-workers identified a green protocol for esterification, which included dimethyl carbonate as the solvent, 2,6-lutidine as a substitute for triethylamine, and Mukaiyama's reagent as a substitute for the carbodiimide such as EDC or DCC (Ref 38, *Green Chem.*, 2021, 23, 6405-6413 DOI: 10.1039/D1GC02251B). We referenced this and updated the text:

Page 3 line 11: “Although the use of EDC and chlorinated solvents create environmental, health and safety (EHS) challenges for monomer synthesis, there are recent coupling protocols(38) employing less toxic reagents and solvents that could be implemented.”

We would also like to direct the reviewer to the data and discussion already presented relating to the synthesis of the isosorbide resin composition using a bulk Fisher esterification, which avoids equimolar, toxic reagents and halogenated solvents for the synthesis of the resin. This is already referenced in the text on page 4 line 8, in Table S1 and Fig. S13-15. Overall, the Fisher esterification produced resins in *ca.* 60% isolated yield with similar purity to those obtained using the EDC protocol. These results suggest that the resin could be manufactured at-scale while adhering to most principles of green chemistry. The Fisher esterification is commonly used to produce bulk commodity chemicals such as acrylates or flavoring agents (DOI: 10.1002/9783527627622.ch8) making it considerably more scalable than Steglich esterification. The text was modified to highlight this aspect:

Page 4 line 4: “...using an acid-catalyzed Fisher esterification in bulk (*i.e.*, without solvent) (see Table S1 and Fig. S13-15), which is commonly used during the manufacture of bulk commodity chemicals.(40)”

The reviewer also brings up a valid concern regarding the use of phosphazene as a depolymerisation catalyst. We screened several additional depolymerisation conditions (varying catalyst, solvent, temperature, additive) and found that DBU and dithiothreitol (DTT) in MeTHF at 80 °C were equally effective (example in Fig S32). DBU is considerably less toxic than the phosphazene, possessing minimal cytotoxicity (*i.e.* good biocompatibility) (*Biomacromolecules* 2015, 16, 2, 507–514 DOI: 10.1021/bm5015443). During our survey of reaction conditions, we also discovered that an un-catalyzed thermal depolymerization in DMF at 140 °C was equally efficient. While DMF is hazardous, this depolymerization process is less wasteful (which is another important principle of green chemistry to consider) and requires less steps so we took this protocol forward. We repeated 2D photoset depolymerizations using this procedure and carried it forward to recycle 3D prints. This new data is now included in the supporting information, summarized in Table S3. For additional data please see: NMR – Fig. S20, S22, S24, S27-29, S33-34, Table S3; HRMS – Fig. S38-46; FTIR – Fig. S65, S67, S69, S72-74; SEC – Fig. S87-88, S90, S92, S95, S96; all data corresponding to recycling prints – Fig. 165-183). The text was amended to reflect:

Page 7 line 15: “While this phosphazene-catalyzed depolymerization was efficient for some formulations, it was not universal and phosphazenes also pose a considerable cytotoxicity hazard(51). Thus, we developed a catalyst-free thermal depolymerization method where the lipoate networks were simply refluxed in DMF for 2 h. We observed that this thermal depolymerization was effective and generally more consistent for the depolymerization of 2D photosets (yields \geq 80%) (Table S3), and equally applicable to the depolymerization of the 3D printed EtLp1:GlyLp3 part (91% yield, 96% cyclic disulfide content).”

The products of depolymerization should be characterized by mass spectroscopy to better understand the compositional differences. NMR is not sufficient to understand the structures formed.

We conducted MS analysis of the depolymerized products obtained from 2D photosets and during the recycling loop for the 3D prints. These data are included in the supporting information (Fig S38-46), although they did not offer any better understanding (compared to the NMR and SEC analysis) of monomer vs polymer composition in the recycled products. The text now includes:

Page 7 line 22: “Mass spectrometry (MS) analysis was also conducted on these depolymerized mixtures but did not offer additional understanding on their composition (Fig. S38-46).”

The mechanical data of the materials should be compared with another classical acrylate used for SLA.

Reviewer 1 made a similar suggestion, and we contextualized the data for our resin against several commercial resins. We direct the reviewer to that previous full response. However, we summarize the text changes here:

Page 6, line 43: “Many of the compositions are already within the range of “soft” commercial resins, such as FormLabs’ Elastic 50A™ or Flexible 80A™ and Photocentrics’ Flexible UV160TR™ (Table S6).”

It would be important to understand the E" profile (by DMTA) of these materials knowing that some formulations were made with 70% crosslinker.

The DMA thermogram of EtLp₁:GlyLp₃ revealed a large rubbery plateau after the glass transition (storage modulus decreased from *ca.* 300 MPa to 2 MPa and held constant) extending from 10–160 °C. Resins containing GlyLp₃ generally possessed wider rubbery plateaus before failure, as compared to IsoLp₂-based resins, with all showing good dimensional stability up to at least 100 °C. DMA data for all material formulations are now included in the manuscript (Figure S103-112) and appropriate discussion in the text has been included. The text was amended to include:

Page 6, line 37: “Analysis of a material composed of EtLp₁:GlyLp₃ revealed a notably large rubbery plateau extending from *ca.* 10–160 °C as assessed by dynamic mechanical analysis (DMA) (Fig. S109). Materials containing GlyLp₃ generally possessed superior dimensional stability (rubbery plateau ≥ 100 °C) as compared to IsoLp₂-based resins (Fig. S103-112), illustrating their large operational window.”

It is mentioned that “ SEC analysis of the recycled resin showed that it had a similar molecular weight profile to the original resin, although a small amount of polymeric product was observed, which indicates that the depolymerization was not 100% effective (Fig. 2G).” Did the authors try to extend the depolymerization time (or change other conditions?) to eliminate polymer residues? A Soxhlet extraction should be performed to evaluate the gel content of the crosslinked and depolymerized material. The products resulting from the extraction should be characterized.

As mentioned in an earlier response, we conducted additional screening of depolymerization reaction conditions to address previous inconsistencies and improve overall efficiency (i.e. increase recovered monomer or cyclic disulfide). Extending the reaction time beyond those that are reported for each method (outlined in Table S2-3 and the experimental section) did not improve the purity (cyclic disulfide content) or increase the isolated yield of the products. The only purification step of the depolymerized products we

performed was to elute the crude mixture through a pad of basic alumina. This is stated in the experimental section for the depolymerization experiments. We do not believe that there is considerable gel content to subsequently remove from the depolymerized resins since they have been passed through alumina (the reaction mixtures were generally highly homogeneous and transparent before this alumina step as well). Thus, these are soluble (oligo)polymeric species that appear (e.g. Fig. 2G or Fig. 4B), as evidenced by their manifestation in SEC analysis. The SEC analysis of all depolymerized products already appears in the supporting information (Fig. S83-99). The reported compositional analysis of the recycled resins is a complete profile of soluble (or extracted) components. Hence, the utility of Soxhlet extraction is limited in this case and would add avoidable waste products to the resin recovery process.

Referee #3 (Remarks to the Author):

This authors describe the synthesis of a platform of new bio-based photopolymer resins for additive manufacturing, followed by depolymerization of printed objects and subsequent re-printing. The manuscript is well-written, concise and of high quality. I agree with the authors that the closed-loop recycling of stereolithographic printed parts is a scientific challenge, and while several research teams attempted to reach that goal, most ended up with a open-loop process, since addition of extra resin components was needed for re-printing. The authors suggest their approach is closed-loop. However, to ensure reliable re-printing, they had to add extra photoinitiator to the recycled resin. Therefore, in my opinion, it is not so different with respect to what they have defined as "open-loop".

We thank the reviewer for their positive comments about this study. With respect to their final comment regarding open-loop vs closed-loop, we are happy to address that where they specifically raise it below.

The following points need attention:

- Page 4 line 79-81: Partial gelation was avoided by reacting both alcohols simultaneously with lipoic acid. What was the reason for the gelation? And why was it circumvented by reacting with both alcohols at the same time? This clearly needs more explanation in the manuscript.

This is partially answered in a related comment from Reviewer 1, comment #5. We direct the reviewer to that response for a more detailed explanation. The primary reason for gelation during cooling of the Fisher esterification reaction is likely due to thermal-induced crosslinking (lipoates have been thermally polymerized by Feringa and co-workers: *Matter* **4**, 1352-1364 (2021) & *Angew. Chem. Int. Ed.* **59**, 5278-5283 (2020). These reports were already referenced in our manuscript, and they have also been added here in the amended text:

Page 4 line 6: "Partial gelation was observed for IsoLp₂ upon cooling of the reaction mixture, which is likely due to thermal-assisted polymerization (41, 42). However, this was avoided when both alcohols were reacted simultaneously with lipoic acid to directly obtain MenLp₁:IsoLp₂ (28:72 wt.%) as the mixed lipoate product was observed to be more stable than either constituent component."

We are still probing the surprising stability enhancement of resins that are formulated (mixed), however this feature seems to be general from our anecdotal observations. As we point out in the updated text below, it may be that each lipoate acts as a diluent for one another (performing a role similar to dilution using

solvent, which itself significantly reduces likelihood of gelation or premature crosslinking). The text was amended:

Page 3 line 17, "...we observed that the mixture was comparatively more stable than either component in isolation (Fig. 2A), possibly a consequence each mixed component serving as a diluent to the other, despite also possessing cyclic disulfide motifs. Nevertheless, it is advantageous for the practical translation of this resin system."

- Page 4 line 91: The pristine resin was formulated without a photoinitiator. Could the authors elaborate on the actual stability of their resin at ambient conditions?

In short, the resin is stable and useable for up to 25 days on the bench-top when protected from light (i.e. stored in an amber bottle). Any deterioration in resin from aging can be addressed by depolymerization back to monomer, effectively regenerating a printable resin. This was also answered in our response to a comment from Reviewer 1, comment #5. We direct the reviewer to that detailed explanation. A summary of the text changes is provided here:

Page 4 line 10: "The formulated resin could be stored on the bench-top when protected from light for several weeks while maintaining a consistent viscosity and rapid photocurability (Fig. S165). It is important to note that aging effects can be neutralized by depolymerizing back to monomer components."

- Page 5 line 116: The depolymerization was not 100% effective. It seems the researchers were able to re-print once, which is already an excellent achievement. However, to what extent is it possible to recycle multiple times (over multiple cycles)?

We have included additional data showing a 2nd re-print of the depolymerized resin, i.e. print, recycle, print, recycle, print, while maintaining adequate resolution during the printing process. This data is now included in Fig. 4, with accompanying discussion, and Fig S165-183. Additional discussion is also offered in the response to the final comment (see below).

- Page 7 line 190: The authors mention a small decrease in curing rate. However, it takes twice as long to reach the gel point for the recycled resin. That seems quite a difference. Please discuss.

We believe there is a strong dependence on the curing and stability of the resin based on the molar concentration of the dithiolane species. Hence, we suggest that the observed difference in curing rate is due to the non-quantitative recovery of the dithiolane species (minor polymer impurities) in the depolymerised resin (96 mol% in 1st recycle and 94 mol% in 2nd recycle) increasing the time to gelation. However, this is easily circumvented by adding a small quantity of photoinitiator, which enables rapid, high-resolution 3D printing. The text was amended to reflect the new data obtained for multiple recycles:

Page 7 line 40: "Photorheological analysis of the EtLp1:GlyLp3 resin samples revealed rapid and consistent curing kinetics (Fig. 4C, Fig. S168). The curing kinetics for the crude recycled resins were slightly slower (Fig. S166-167), however this can be rectified by adding photoinitiator after which all resins reached a gel point within 1.4 s. The small decrease in curing rates is likely attributed to the minor oligomer impurity and its effect on the absorption profile of the recycled resins. Nevertheless, the three cured resins possessed

similar plateau moduli, indicating negligible effects on broader bulk material properties from adding photoinitiator to the formulation (Fig. 4C).”

- Page 7 line 192: In the introduction, the authors refer to open-loop processes that need addition of extra resin components for re-printing (and re-curing). They claim their process is, in contrast to others, closed-loop (Figure 1C). However, to ensure reliable re-printing, they add a small quantity of photoinitiator to the recycled resin, while this was not needed for the first cycle. From that viewpoint, to what extent is this a real closed-loop process?

We appreciate the authors viewpoint on this aspect of the manuscript, and it opens perhaps a more philosophical question rather than a technical one. A truly closed-loop process would require *quantitative regeneration* of the original resin from the previous one and *no exchange of any matter* externally. This is unrealistic for *any* material recycling process. Even the usage of catalysts and solvents which are not regenerated during the process would thus afford an un-ideal “closed loop” process, more broadly speaking. A better description for this system is perhaps provided by another reviewer (see reviewer 1 summary) who refers to this as the “most recyclable photocurable resin to date” which we think is an accurate description. And given that our resin does satisfy most, if not all, of the requirements for a circular recyclable photocurable resin, we believe this represents the most “closed-loop” cycle currently available and thus it is reasonable to refer to it as such. We offer further support for our claim in the following paragraphs.

The current state-of-the-art in recyclable photoresins is clearly open-loop (Ref 23-24) and distinct from our process. In these recent approaches there are several hallmarks of this open-loop classification: 1) depolymerization requires an external reagent e.g. excess thiols to facilitate thiol–thioester (Ref 23) or thiourethane exchange (Ref 24) and 2) additional reactive diluent e.g. thioester diallyl ether (Ref 23) or isocyanate and solvent (Ref 24) is required to provide a curable re-formulated mixture. For example, in Ref 24 (which is cutting edge in recyclable 3D printing – published on Sept 7, 2023), to depolymerize the prints “2 equivalent excess of the triSH (or 19.5 wt%) in acetone and 0.1 wt% of 1,1,3,3-tetramethylguanidine (TMG) as basic catalyst” were added. The recovered resin (95% yield) was markedly different than the initial formulation (viscosity increased by ~ 1 order of magnitude) since it was incompletely depolymerized to only yield oligomers (M_n 1st recycle = 11.7 kDa vs 5.4 kDa for the initial component. Finally, to re-print, the components were added as follows: “The recycled oligomers (2.95 mmol thiol [SH]) were recross-linked by adding stoichiometric amounts of PPG-diIPDI (M_n : 2444.6 g/mol) (2.95 mmol isocyanate [NCO]), 0.5 wt% of TBD·HBPh₄ and 0.25 wt% ITX dissolved in 30 wt% of acetone according to the complete system similarly to the original formulation.” We estimate the recycled resin to incorporate only ~46% by mass of the resin components from the previous loop, a second cycle was not reported.

The processes in these prior examples are clearly different when compared to adding a small quantity of photoinitiator (≤ 2.5 wt.%). While we do not observe quantitative depolymerization (~95% recovery of cyclic disulfides, > 90% yield), we avoid adding “extra” monomer or reactive/non-reactive diluent to re-print the recycled resins over multiple cycles – everything that results from the depolymerisation is included in the recycled resin and only a small quantity of photoinitiator is added –at the first recycle, our recycled resin contains 98.5% by mass of the resin components from the previous loop (97.5% at the second recycle).

Moreover, our recycling method can now proceed via an uncatalyzed, thermal depolymerization method and eliminates catalyst components, which are required to depolymerize prints in Ref 24. To address any concern about the total number of cycle loops, and need for additional photoinitiator, we developed another

“looped” recycling path for any deteriorated lipoate resin (due to aging and/or successive recycling). We conducted a hydrolytic depolymerization (NaOH in H₂O/DMF mixture) of the lipoate resin to depolymerize the polydisulfide (monomer regeneration) and hydrolyze the lipoate ester. This yielded alcohol and lipoic acid, i.e. the original feedstock components. Clearly, these are the same starting materials from which our monomers are made and could be transformed back to monomer in one-step. We include several cursory experiments related to this aim to establish feasibility (Fig S35-37 - NMR; Fig. S75-77 – FTIR; Fig. S97-99 – SEC Table S4 – summary of experiments). We have added this discussion to the last section when discussing print recycling:

Page 8, line 8: “To address any aging and/or successive recycling effects on the resin, we also established a further potentially infinite “looped” recycling pathway. To this end, hydrolytic depolymerization (NaOH in H₂O/DMF) of the resin was employed to regenerate the original resin components, namely alcohols and lipoic acid, (Table S4, Fig. S35-37, S75, S97-99) which could be easily integrated back into the material cycle to form virgin resin.”

Reviewer Reports on the First Revision:

Referees' comments:

Referee #1 (Remarks to the Author):

Dove, Worch, and co-workers provide a comprehensive rebuttal, including extensive revisions to the originally submitted manuscript that address many of the reviewer's concerns, and overall strengthen the work. I personally do not agree with the concerns raised by reviewer 3 regarding this process not being "closed-loop" due to the minor addition of photoinitiator – the authors do a good job rationalizing the use of their terminology. I do agree that the concerns raised by reviewer 2 with the use of phosphazene, particularly in the quantities that are used. Their alternative depolymerization methods provided upon revision are compelling, however phosphazene is still highlighted in the main manuscript (Figure 2 and associated text). Bringing up the concerns with and alternatives to phosphazene earlier is recommended.

With regard to mechanical property optimization – the authors reasonably note that this has been optimized for ~decades for acrylic systems to provide high performance materials from stiff and strong to soft and elastic. While I agree that it is unreasonable to expect that this would be shown for a new system (lipoates) in a single report, it is reasonable (in my view) to provide compelling suggestions on how one could realistically envision getting there and what hurdles may be encountered. The authors provide some suggestions in their rebuttal, such as incorporating hydrogen bonding, however, I remain skeptical that networks with high disulfide content will be able to provide competitive thermomechanical properties relative to acrylics. My concerns stem from the intrinsic nature of disulfide bonds being relatively flexible and weak in comparison to carbon-carbon bonds, which of course is the reason for their reprocessability. While I am happy to be wrong in this skepticism, the authors are challenged to provide stronger arguments to this point than what is currently given.

The new thermal characterization showing good stability of the cured objects up to ~200 C was impressive, and raises the potential impact of the work. However, it was also surprising given that heating in DMF at 140 C is sufficient for near quantitative depolymerization. Likely degradation of DMF that gives off dimethylamine is the cause of this lower depolymerization temperature, however the authors are encouraged to discuss this discrepancy in depolymerization vs. degradation temperature further. Additionally, while Td5 values of ~200+ C were found with TGA, many of the DMA traces drop off at temperature at or below 150 C. The authors are encouraged to address this difference as well.

Overall, this is a solid report that will draw attention from the materials community. However, I still express concerns regarding the novelty and impact that I provided in my initial review, which I only emphasize here given the consideration for publication in Nature as a high-profile journal. One component that could strengthen the novelty is a more clear description of the mechanistic elements that are unique in the depolymerization process herein that make it more efficient than what has previously been reported for other lipoate polymers. The authors emphasize the lack of examples

showing efficient depolymerization, but the process they use seems quite similar to others tried on non-3D printed disulfide polymers. While I am not suggesting a full mechanistic study be done, I think more emphasis in the main text on the novelty of their process (i.e., where it stems from) that can be extended to other lipoate (or disulfide) polymers would provide more of a generalizable strategy that would be of broad utility, and thus raise the potential impact of this work.

1. A stylistic suggestion that the authors may consider: I think that Figure 4A in the revised manuscript takes up a lot of real estate, but the information it provides is comparable to the other parts of that figure (B, C, D, etc.). I suggest overlaying the three NMR spectra and zoom into a characteristic region to showcase the similarities between the recycled resins and the virgin resin. The full spectra could then be provided in the SI and referenced in the main text.

Referee #2 (Remarks to the Author):

The authors have properly addressed the comments made in my first revision, so that in my opinion the manuscript can be accepted in its present form.

Referee #3 (Remarks to the Author):

The points raised in the previous round of review have been satisfactorily addressed.

Author Rebuttals to First Revision:

Referees' comments:

Referee #1 (Remarks to the Author):

Dove, Worch, and co-workers provide a comprehensive rebuttal, including extensive revisions to the originally submitted manuscript that address many of the reviewer's concerns, and overall strengthen the work. I personally do not agree with the concerns raised by reviewer 3 regarding this process not being "closed-loop" due to the minor addition of photoinitiator – the authors do a good job rationalizing the use of their terminology. I do agree that the concerns raised by reviewer 2 with the use of phosphazene, particularly in the quantities that are used. Their alternative depolymerization methods provided upon revision are compelling, however phosphazene is still highlighted in the main manuscript (Figure 2 and associated text). Bringing up the concerns with and alternatives to phosphazene earlier is recommended.

We thank the reviewer for their positive comments about the revised manuscript. We recognise that the highlighting of the phosphazene route early in the manuscript remains, this is a consequence of that being how that portion of the work was conducted. We felt it would be dishonest to change that substantially and prefer to retain the order of results as presented. We do agree however that it is important to note the advances earlier in the manuscript and highlight the 'greener' methods that we have developed. To that end, we have:

- 1) Removed the specificity to phosphazene in Figure 2 to make the scheme more generic
- 2) Added a sentence to the main text, where the phosphazene-mediated depolymerisation is first mentioned to highlight the work reported later in the manuscript: "Other greener depolymerization methods were also screened and found to be effective (*vide infra*)."

With regard to mechanical property optimization – the authors reasonably note that this has been optimized for ~decades for acrylic systems to provide high performance materials from stiff and strong to soft and elastic. While I agree that it is unreasonable to expect that this would be shown for a new system (lipoates) in a single report, it is reasonable (in my view) to provide compelling suggestions on how one could realistically envision getting there and what hurdles may be encountered. The authors provide some suggestions in their rebuttal, such as incorporating hydrogen bonding, however, I remain skeptical that networks with high disulfide content will be able to provide competitive thermomechanical properties relative to acrylics. My concerns stem from the intrinsic nature of disulfide bonds being relatively flexible and weak in comparison to carbon-carbon bonds, which of course is the reason for their reprocessability. While I am happy to be wrong in this skepticism, the authors are challenged to provide stronger arguments to this point than what is currently given.

We thank the reviewer for their response and comment. They are correct that achieving materials through this route to compete with acrylic resins will be a considerable challenge. As they recognise though, this would be a step too far to include in this report alongside the closed loop recycling that is the focus of the manuscript. We do believe that there is optimism to significantly stiffen the materials through judicious choice of chemistries with our focus now being on introducing second networks into the system through non-covalent interactions – other routes may be possible too. We particularly highlight the work of Feringa and coworkers (Y. Deng et al., Acylhydrazine-based reticular hydrogen bonds enable robust, tough, and dynamic supramolecular materials. *Sci. Adv.* 8, eabk3286 (2022)) in which the synthesis of acylhydrazine-functional lipoates led to a significant increase in mechanical strength. We believe that harnessing this approach, perhaps through a combination of multiple non-covalent networks, provides the potential to shift the dial further on this technology. We will not know its limits until we try!

To strengthen the manuscript text in this regard we have edited the section as follows to make this clearer and add in other strategies that have been investigated in other systems to provide strong non-covalent networks from which we are drawing inspiration:

“Our current efforts to further increase mechanical performance are focused on combining the inherent flexibility endowed by their low T_gs with network reinforcement through non-covalent interactions such as metal coordination strategies(37), stereochemical effects(41) or hydrogen bonding groups(42, 43). The use of acylhydrazines(44), which introduce strong hydrogen bonding interactions, has yielded lipoate networks with UTS near 50 MPa and Young’s modulus up to 340 MPa, thus demonstrating that significant stiffening of the lipoate resin platform could be achievable to compete more broadly with state-of-the-art commercial resins.”

The new thermal characterization showing good stability of the cured objects up to ~200 C was impressive, and raises the potential impact of the work. However, it was also surprising given that heating in DMF at 140 C is sufficient for near quantitative depolymerization. Likely degradation of DMF that gives off dimethylamine is the cause of this lower depolymerization temperature, however the authors are encouraged to discuss this discrepancy in depolymerization vs. degradation temperature further. Additionally, while T_{d5} values of ~200+ C were found with TGA, many of the DMA traces drop off at temperature at or below 150 C. The authors are encouraged to address this difference as well.

We are happy to provide more details on the solvent-mediated depolymerisation process. As now noted in the manuscript text, we were inspired by the work of Odelius and coworkers (L. Cederholm et al., “Like Recycles Like”: Selective Ring-Closing Depolymerization of Poly(L-Lactic Acid) to L-Lactide. *Angew. Chem. Int. Ed.* 61, e202204531 (2022)) in which they reported that the ceiling temperature of polylactide could be reduced by judicious solvent selection in which to undertake depolymerisation reactions. They hypothesise that this was a result of favourable solvent interactions on the monomer-polymer equilibrium thus helping to favour monomer, and thus resulting in depolymerisation at lower temperatures. In our search for improved depolymerisation conditions, in the absence of toxic catalysts, we decided to evaluate

this idea in application to our lipoate networks and were pleased to see that it was applicable to the ring-chain equilibrium present. To this end, we have amended a sentence to now read:

“While this phosphazene-catalyzed depolymerization was efficient for some formulations, it was not universal and phosphazenes also pose a considerable cytotoxicity hazard(45). Thus, inspired by the work of Odelius and coworkers in which the ceiling temperature of polylactide was reduced by favorable solvent interactions on the monomer-polymer equilibrium,(46) we developed a catalyst-free thermal depolymerization method where the lipoate networks were simply refluxed in DMF for 2 h.”

We are also happy to provide more context on the DMA thermograms in relation to other thermal properties. The DMA “drop-off” indicates bulk mechanical failure of the sample, most likely due to flow as the temperature approaches the network flow activation energy. In conventional covalent adaptable networks, bulk material flow before degradation is a universal feature and it is not unreasonable to observe similar behaviour in our materials since they are networks with dynamic covalent crosslinks. Moreover, depolymerization (i.e. ring-closing depolymerization that could lead to volatile species captured by TGA analysis) is only one possible mechanistic pathway, which can be enhanced by the effect of solvent on the thermodynamics of the polymer network (as highlighted above in the DMF depolymerisation discussion). In effect, we do not believe that the material flow in DMA thermograms is due to depolymerisation of the sample. Instead, disulfide metathesis or exchange is expected to be a competing reaction, as proposed in other lipoate systems (Zhang & Waymouth, “1,2-Dithiolane-Derived Dynamic, Covalent Materials: Cooperative Self-Assembly and Reversible Cross-Linking” *J. Am. Chem. Soc.* 139, 3822–3833 (2017); Alraddadi et al., “Renewable and recyclable covalent adaptable networks based on bio-derived lipoic acid” *Polym. Chem.* 12, 5796 (2021)). In fact, in the latter report DMA analysis shows bulk material flow at around 150 °C (Fig. 5b) while the degradation temperatures are >200 °C. We have referenced this report at the end of the DMA discussion:

“Analysis of a material composed of EtLp1:GlyLp3 revealed a notably large rubbery plateau extending from ca.10–160 °C as assessed by dynamic mechanical analysis (DMA) (Fig. S109), which is similar to previous lipoate networks(32).”

Overall, this is a solid report that will draw attention from the materials community. However, I still express concerns regarding the novelty and impact that I provided in my initial review, which I only emphasize here given the consideration for publication in *Nature* as a high-profile journal. One component that could strengthen the novelty is a more clear description of the mechanistic elements that are unique in the depolymerization process herein that make it more efficient than what has previously been reported for other lipoate polymers. The authors emphasize the lack of examples showing efficient depolymerization, but the process they use seems quite similar to others tried on non-3D printed disulfide polymers. While I am not suggesting a full mechanistic study be done, I think more emphasis in the main text on the novelty of their process (i.e., where it stems from) that can be extended to other lipoate (or disulfide) polymers would provide more

of a generalizable strategy that would be of broad utility, and thus raise the potential impact of this work.

We hope that we have addressed the reviewers comment adequately in our responses above. We believe that this is a significant advance in the state of the art with respect to closed loop 3D printing, and that some of the other methodologies reported will benefit the wider communities researching related areas.

1. A stylistic suggestion that the authors may consider: I think that Figure 4A in the revised manuscript takes up a lot of real estate, but the information it provides is comparable to the other parts of that figure (B, C, D, etc.). I suggest overlaying the three NMR spectra and zoom into a characteristic region to showcase the similarities between the recycled resins and the virgin resin. The full spectra could then be provided in the SI and referenced in the main text.

We thank the reviewer for their comments and have made changes to Figure 4 in line with their suggestion. The full-width NMR spectra with associated integrations are now included as a stacked plot in Fig. S34.

Referee #2 (Remarks to the Author):

The authors have properly addressed the comments made in my first revision, so that in my opinion the manuscript can be accepted in its present form.

We are grateful for the supportive comments of the reviewer.

Referee #3 (Remarks to the Author):

The points raised in the previous round of review have been satisfactorily addressed.

We are grateful for the supportive comments of the reviewer.